# Top-down machine learning approach for high-throughput single-molecule analysis

David S White[1,2], Marcel P Goldschen-Ohm[3], Randall H Goldsmith[2]*, Baron Chanda[1,4]†*

[1]Department of Neuroscience, University of Wisconsin-Madison, Madison, United States; [2]Department of Chemistry, University of Wisconsin-Madison, Madison, United States; [3]Department of Neuroscience, University of Texas at Austin, Austin, United States; [4]Department of Biomolecular Chemistry University of Wisconsin-Madison, Madison, United States

**Abstract** Single-molecule approaches provide enormous insight into the dynamics of biomolecules, but adequately sampling distributions of states and events often requires extensive sampling. Although emerging experimental techniques can generate such large datasets, existing analysis tools are not suitable to process the large volume of data obtained in high-throughput paradigms. Here, we present a new analysis platform (DISC) that accelerates unsupervised analysis of single-molecule trajectories. By merging model-free statistical learning with the Viterbi algorithm, DISC idealizes single-molecule trajectories up to three orders of magnitude faster with improved accuracy compared to other commonly used algorithms. Further, we demonstrate the utility of DISC algorithm to probe cooperativity between multiple binding events in the cyclic nucleotide binding domains of HCN pacemaker channel. Given the flexible and efficient nature of DISC, we anticipate it will be a powerful tool for unsupervised processing of high-throughput data across a range of single-molecule experiments.

*For correspondence:
rhg@chem.wisc.edu (RHG);
chanda@wisc.edu (BC)

Present address: †Department of Anesthesiology, Washington University School of Medicine, St. Louis, United States

## Introduction

Single-molecule methods are powerful tools for providing insight into heterogeneous dynamics underlying chemical and biological processes otherwise obscured in bulk-averaged measurements (*Moerner et al., 2015*). Use of these techniques has expanded rapidly, with modalities spanning electrophysiology, fluorescence, and force spectroscopy to probe diverse physical phenomena. Generally, single-molecule data are obtained as a time trajectory where molecular behavior is observed as a series of transitions between a set of discrete states obscured by experimental noise. Following the growing realization that molecules involved in physiological and chemical processes exhibit complex kinetics and a diversity of behavior, there is an increasing demand for high-throughput technologies to adequately sample different sub-populations and rare but important events (*Hill et al., 2017*). As a result, there has been tremendous progress in improving both the number of single molecules that can be observed simultaneously and the total observation time of each molecule. For example, the observation window prior to photobleaching in conventional fluorescence paradigms such as single-molecule Förster resonance energy transfer (smFRET) or colocalization single-molecule spectroscopy (CoSMoS) can be dramatically extended with recently developed photostable dyes (*Grimm et al., 2015*; *Altman et al., 2012*). The current generation of metal-oxide semiconductor (sCMOS) detectors enables simultaneous imaging of $1 \times 10^4$ molecules in a total internal fluorescence microscopy (TIRFM) configuration and can be coupled with nanofabricated zero-mode waveguides (ZMWs) to enable access to high concentrations (*Levene et al., 2003*; *Chen et al., 2014*; *Juette et al., 2016*). Non-fluorescence-based single-molecule experiments such as plasmon rulers, scattering, magnetic tweezers, and single-molecule centrifugation generate a tremendous amount

**eLife digest** During a chemical or biological process, a molecule may transition through a series of states, many of which are rare or short-lived. Advances in technology have made it easier to detect these states by gathering large amounts of data on individual molecules. However, the increasing size of these datasets has put a strain on the algorithms and software used to identify different molecular states.

Now, White et al. have developed a new algorithm called DISC which overcomes this technical limitation. Unlike most other algorithms, DISC requires minimal input from the user and uses a new method to group the data into categories that represent distinct molecular states. Although this new approach produces a similar end-result, it reaches this conclusion much faster than more commonly used algorithms.

To test the effectiveness of the algorithm, White et al. studied how individual molecules of a chemical known as cAMP bind to parts of proteins called cyclic nucleotide binding domains (or CNDBs for short). A fluorescent tag was attached to single molecules of cAMP and data were collected on the behavior of each molecule. Previous evidence suggested that when four CNDBs join together to form a so-called tetramer complex, this affects the binding of cAMP. Using the DISC system, White et al. showed that individual cAMP molecules interact with all four domains in a similar way, suggesting that the binding of cAMP is not impacted by the formation of a tetramer complex.

Analyzing this data took DISC less than 20 minutes compared to existing algorithms which took anywhere between four hours and two weeks to complete. The enhanced speed of the DISC algorithm could make it easier to analyze much larger datasets from other techniques in addition to fluorescence. This means that a greater number of states can be sampled, providing a deeper insight into the inner workings of biological and chemical processes.

of data through parallel measurement of hundreds of molecules with orders of magnitude longer recordings than a typical fluorescence experiment (*Berghuis et al., 2016*; *Ye et al., 2018*; *Yang et al., 2016*; *Popa et al., 2016*; *Young and Kukura, 2019*).

Unfortunately, despite these advances in generating statistically robust data sets, standard analysis algorithms impose a computational bottleneck at this scale of data generation (*Hill et al., 2017*; *Juette et al., 2016*). This is particularly true when the dynamics and physical states of a system are unknown.

Typical statistical modeling of single-molecule trajectories often adopts one of two approaches. The first is a probabilistic approach that models a molecule's behavior as a Markov chain, wherein the molecule transitions between hidden discrete states whose outputs are measured experimentally (hidden Markov model, HMM). This involves estimating the transition probabilities between a small set of postulated states with defined outputs using methods to maximize the likelihood of the model given the observations or Bayesian inference to estimate model parameter distributions. Numerous software packages have been developed for implementing HMMs, such as QuB (*Nicolai and Sachs, 2013*; *Qin et al., 2000*), HaMMy (*McKinney et al., 2006*), SMART (*Greenfeld et al., 2012*), vbFRET (*Bronson et al., 2009*), ebFRET (*van de Meent et al., 2014*) and SPARTAN (*Juette et al., 2016*), each of which utilize a different HMM training method. For example, QuB implements the fast segmental k-means algorithm (SKM) which combines k-means clustering and the Viterbi algorithm to identify transitions between postulated states (*Juang and Rabiner, 1990*), whereas vbFRET adapts variational Bayesian inference for parameter estimation at faster speeds than traditional HMM training in both smFRET and single-particle tracking experiments (*Blanco and Walter, 2010*; *Persson et al., 2013*). Although powerful statistical tools are very useful for single-molecule analysis, HMMs have notable limitations, especially in the context of high-throughput analysis and unknown system dynamics. For example, HMMs are often used in a supervised manner where the user postulates model parameters such as the number of states, their measured outputs, and the allowed transitions between them. As this information is often not known a priori, it is desirable to test multiple models and rank them according to Bayesian probabilistic approaches or objective functions, such as the Bayesian Information Criterion (BIC). This process can dramatically increase the analysis time

to ensure the parameters space has been sufficiently explored, which restrict their usefulness in high-throughput single-molecule analysis. Although variants such as infinite HMMs using Bayesian nonparametric inference try to naturally learn the trajectory of the states without the typical parametric model selection, these too are often computationally prohibitive for large data sets (*Hines et al., 2015*; *Sgouralis and Pressé, 2017*; *Sgouralis et al., 2018*).

The second class of single-molecule analysis approaches idealization as an unsupervised clustering problem from machine learning (*Li and Yang, 2019*). Typically, clusters of intensity values (e.g. states) are determined using bottom-up hierarchical agglomerative clustering (HAC) algorithms, which begin by treating each observation of $N$ total observations as singleton clusters and perform $N$-1 iterations wherein pairs of clusters are merged until all data-points belong to a single cluster. For each number of possible clusters, an objective function can be minimized to find the optimal trade-off between the complexity and fit. This implementation results in time complexity of $O(N^2)$ owning to the need of computing a $N$x$N$ similarity matrix to determine which clusters should be merged at each iteration. In practice, a separate algorithm called change-point (CP) detection precedes HAC to reduce the solution domain of the objective function by identifying statistically significant stepwise changes in signal over time. We denote this combination of algorithms as CP-HAC. At each identified CP, the data are divided into two segments, each described by the mean values of the data-points between sequential change-points. By using the segments as initial clusters rather than all $N$ data-points, the HAC computation can be dramatically reduced. The pioneering application of CP-HAC to single-molecule data addressed CP detection and clustering in the presence of Poisson noise (*Watkins and Yang, 2005*). Variants of this framework such as STaSI use other merit functions for Gaussian noise, including the Student's t-test for fast CP detection and minimum description length for state selection (*Shuang et al., 2014*). An advantage of CP-HAC methods is that they only require a confidence interval and/or an objective function for the analysis, unlike HMMs which require a model to fit. This makes them very attractive in situations where there is no prior knowledge about the different physical states. In common experimental modalities such as smFRET, CP-HAC methods offer superior computational speed over HMM approaches; however, their quadratic time-complexity renders them inefficient on long trajectories (*Shuang et al., 2014*). In addition, simulation studies have suggested CP-HAC algorithms yield lower event detection accuracy than HMM approaches (*Hadzic et al., 2018*).

Despite the utility of HMM and CP-HAC methods, there is an outstanding need for an analysis platform to provide accurate model-free idealization with sufficiently high computational performance to keep up with the increasing scale of data generation. Although advances in computing hardware can, to a degree, mitigate these issues (*Smith et al., 2019*; *Song and Yang, 2017*), there remains a pressing need for more computationally efficient algorithms. Here, we present a new algorithm for efficient and accurate idealization of large single-molecule datasets in a model-independent manner. Our method, DISC (DIvisive Segmentation and Clustering), enhances existing statistical learning methods and enables rapid state and event detection. The DISC algorithm draws inspiration from other model-free algorithms like CP-HAC and SKM that rely on unsupervised algorithms, such as k-means and hierarchical clustering. We advance these ideas by adapting divisive clustering algorithms from data mining and information theory to improve the rate and accuracy of identifying signal amplitude clusters (states) in a top-down process as opposed to the typical bottom-up clustering (*Pelleg and Moore, 2000*; *Karypis et al., 2000*; *Hamerly and Elkan, 2003*). We further couple our model-free state detection with the Viterbi algorithm to enable robust event detection on par with HMM methods (*Juang and Rabiner, 1990*; *Qin, 2004*; *Viterbi, 1967*). Overall, DISC is an unsupervised method that combines statistical learning approaches with the high event detection accuracy of HMMs at a fraction of the computational cost, enabling convenient application to large datasets.

## Theory

### Motivation

The goal of the DISC algorithm is time series idealization: the hard assignment of data points into discrete states. DISC approaches the problem of idealization as an unsupervised problem in machine learning wherein the number of significant states for a given single-molecule trajectory are not known a priori. This process of learning both the significant states and the transitions between them

is accomplished in three phases: 1) divisive segmentation, 2) HAC, and 3) the Viterbi algorithm (*Figure 1a*, *Figure 1—figure supplement 1*). The first two phases use unsupervised statistical learning to identify the intensities of states following an appropriate user-specified objective function. The second phase uses the Viterbi algorithm to decode the most probable sequence of transitions between the identified states.

The primary distinction of DISC vs other model-free single-molecule analysis algorithms is the use of divisive clustering. Rather than being limited by the quadratic time-complexity of bottom-up HAC, DISC adopts the exact opposite hierarchical clustering algorithm: top-down divisive clustering. Divisive clustering initializes all data-points to a single cluster and iteratively splits data into sub-clusters until each data point is its own singleton cluster. Given that there are $2^{N-1}-1$ ways of spitting $N$ data points into two sub-clusters, the complexity of top-down processes has led to their infrequent use. However, efficient implementations involving the use of sub-routines to determine how clusters should be split and whether the new clusters should be accepted has resulted in more efficient and accurate algorithms. For example, clusters can be sub-partitioned using k-means clustering and an objective function can determine if the fit of the data improves with an additional sub-cluster

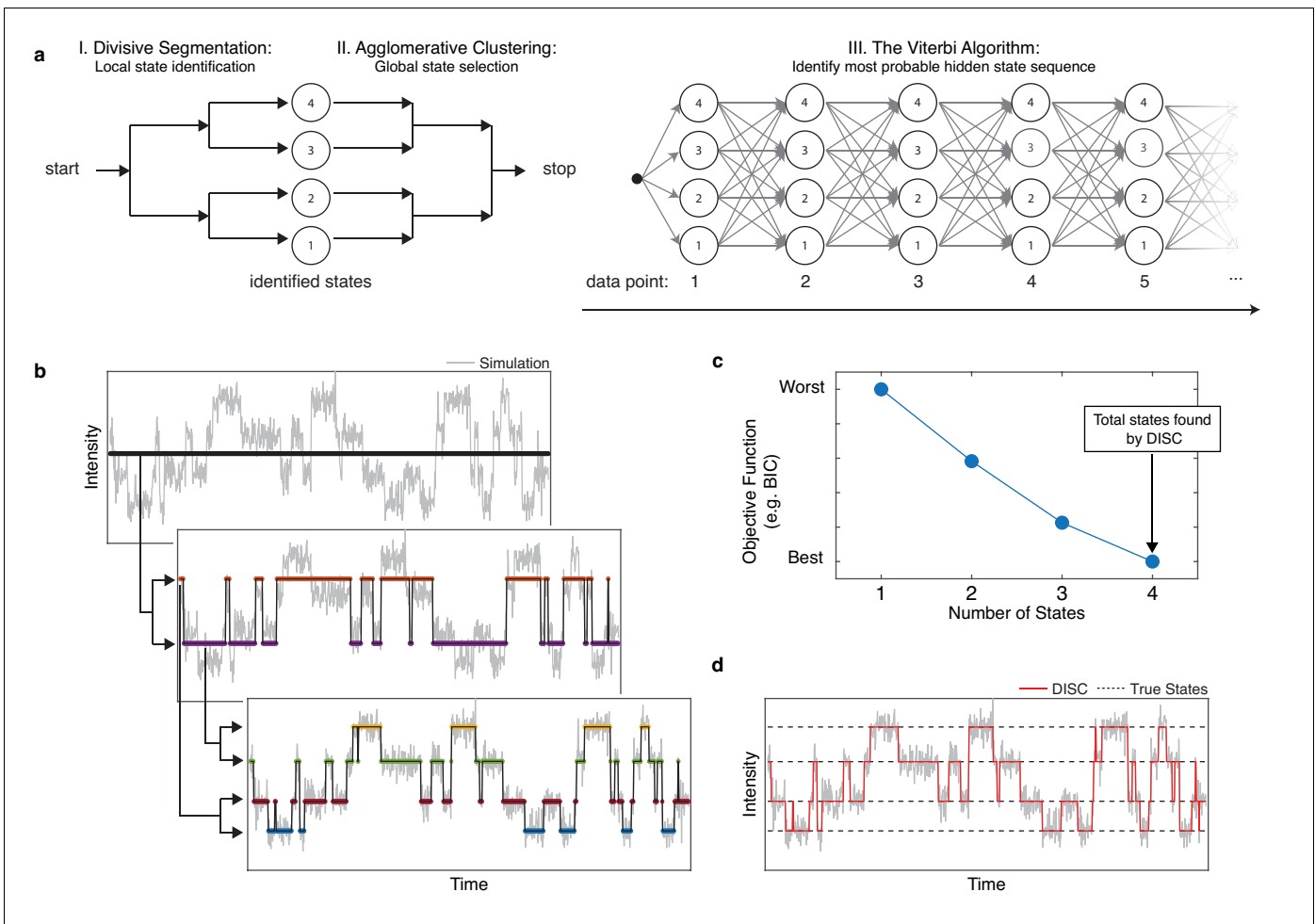

**Figure 1.** Overview of DISC. (**a**) The major steps of the DISC algorithm combining unsupervised statistical learning with the Viterbi algorithm. (**b**) Stepwise discovery of states locally through divisive segmentation on a simulated trajectory. (**c**) HAC iteratively groups identified states to minimize an objective function for the fit of the whole trajectory to avoid overfitting. (**d**) The Viterbi algorithm is applied to identify the most probable hidden state sequence. The final fit by DISC (red) is overlaid against the true states in the simulation (dashed).

The online version of this article includes the following source data and figure supplement(s) for figure 1:

**Source data 1.** Plotted simulated data with DISC fit.
**Figure supplement 1.** Workflow of DISC.

(*Pelleg and Moore, 2000*; *Karypis et al., 2000*; *Hamerly and Elkan, 2003*). Therefore, unlike HAC which makes a final decision at the end of the clustering, top-down processes make a binary decision during each iteration. For each iteration, if the objective function does not improve with the addition of new sub-clusters, the split is rejected. If the fit does improve, the split is accepted and each sub-cluster is again further partitioned into two new sub-clusters. The algorithm terminates once clusters cannot be split for further improvement of the objective function. Given this internal heuristic of cluster acceptance or rejection, the most probable number of states is typically identified within a few iterations, as opposed to the *N*-1 iterations necessary for HAC. In addition, when using k-means for partitioning data points into sub-clusters, the top-down algorithms can reach O(*N*) time complexity (*Karypis et al., 2000*). This simple implementation and accelerated computational speed make divisive clustering an attractive alternative that enables high-throughput single-molecule idealization.

## Divisive segmentation

The first phase of DISC is divisive segmentation. Consider an observed single-molecule trajectory $x = \{x_1, \ldots x_N\}$ where each $x_n$ is the observed intensity value $x$ at time-stamp $n$ for $N$ total observations contaminated by Gaussian noise. Like CP-HAC, the goal of the divisive segmentation is to identify and allocate each data-point into the optimal number of idealized states denoted by $K$. Following our Gaussian assumption, each state $\phi_j \in \{\phi_1, \ldots, \phi_K\}$ is described by the mean ($\mu$) and standard deviation ($\sigma$) of data points allocated to the state $\phi_j = (\mu_j, \sigma_j)$. We denote a series of transitions between states as $y = \{y_1, \ldots y_N\}$ where $y_n \in \{\phi_1, \ldots, \phi_K\}$ and $1 \leq K \leq N$. Divisive segmentation aims to iteratively yield $x$ by determining whether data-points in a given cluster are better described by one or two states. At the onset of divisive segmentation, it is assumed that $x$ is described by a single idealized state. We will denote this initial fit as $y_0 = \{y_{0_1}, \ldots y_{0_N}\}$ where $y_{0_i} \in \{\phi_0\}$ and $\phi_0 = (\mu_0, \sigma_0)$.

Allocating each data-point into two unique states is accomplished in two sequential phases: CP detection and k-means clustering. As opposed to standard divisive algorithms that allocate data points via k-means clustering only, we find CP identification prior to clustering advantageous. Not only does it reduce the solution domain of the objective function, but it also speeds up subsequent clustering while providing a reasonable estimate of state transitions. CP detection is performed with the popular recursive binary segmentation algorithm (*Watkins and Yang, 2005*; *Scott and Knott, 1974*). For each time stamp $n$ in $x$, a hypothesis test is conducted to evaluate the probability that a CP occurred at position $n$ via

- $H_0$: a CP did not occur at position $n$
- $H_1$: a CP did occur as position $n$

In the context of single-molecule idealization, a CP is the location indicating a significant difference in mean intensity values between the data segments separated at location $n$, where the mean values of each segment are computed by

$$\mu_1 = \frac{1}{n}\sum_{i=1}^{n} x_i \qquad \mu_2 = \frac{1}{N-n}\sum_{i=n+1}^{N} x_i \tag{1}$$

To determine whether there is a statistically significant difference between the two segments, we use a two-way Student's t-test of unequal sample size but uniform variance to evaluate the differences in mean. This is the same approach used in STaSI (*Shuang et al., 2014*). Specifically, a *t*-value is computed for each position $n$ by

$$t_k = \frac{|\mu_1 - \mu_2|}{\sigma\sqrt{\frac{1}{n} + \frac{1}{N-n}}} \tag{2}$$

where $\sigma$ is the estimated standard deviation of uniform noise (*Shuang et al., 2014*). The most probable CP location $c$ corresponds to the maximum t-value $t_{max}$ given by

$$c = \underset{k}{argmax}\, k(t) \tag{3}$$

$$t_{max} = \max_k k(t) \tag{4}$$

For a user specified confidence interval, a critical value is used to determine whether to accept the change-point. If $t_{max}$ > critical-value, we reject $H_0$ and the CP is accepted. This in turn segments the data at position $c$. As there are likely multiple CPs in $x$, the algorithm continues in a recursive manner by searching within each new segment $s_1 = \{x_1, \dots x_c\}$ and $s_2 = \{x_{c+1}, \dots x_N\}$. This process terminates when no significant changes in mean intensity are found within any segment. Importantly, the confidence interval set by the user plays a crucial role by acting as a hard threshold for false positive rate.

Following the completion of CP detection, times-series $x2$ can be described as a series of $C+1$ intensity segments where $C$ is the total number of CP identified given by an idealized state trajectory where $y_n 2 \in \{\phi_1, \dots, \phi_K\}2$ and $1 \leq K \leq C+1$. Like CP-HAC, the next goal is to discover the optimal number of states $K$ into which to cluster the $C+1$ intensity segments generated by CP detection. Rather than iteratively merging each segment like bottom-up algorithms, we use divisive segmentation to cluster the data points in a top-down fashion. For divisive segmentation, all identified segments are partitioned into two unique clusters using the k-means algorithm, where the center of each cluster is described by the mean values of the CP-idealized data points within the cluster. For efficiency, DISC uses a modified k-means algorithm that is both deterministic and faster than standard implementations through use of triangle inequality for computational reduction (*Elkan, 2003*). Overall, this results in a series of transitions between two states $y_1 = \{y_{1_1}, \dots y_{1_N}\}2$, where $y_{1_n} 2 \in \{\phi 2_1, \phi 2_2\}$ and $\phi_j = (\mu_j, \sigma_j)2$ the corresponds to a state assignment for each observation in $x2$.

Now that the data-points are allocated to two separate clusters with identified transitions, the goal is to determine if $x$ is better fit with one or two states ($y_0$ vs $y_1$). Like CP detection, this decision follows a hypothesis test where

- $H_0$: the data corresponds to one unique state
- $H_1$: the data corresponds to two unique states

To determine whether one or two states provides a better fit, we use the Bayesian Information Criterion (BIC) which is defined in a general form as

$$BIC = -2\ln(\hat{\mathcal{L}}) + M\,ln(N) \tag{5}$$

where $\hat{\mathcal{L}}$ is the likelihood for the estimated model with $M$ free parameters (*Schwarz, 1978*). The likelihood that the observations $x$ arose from a single state ($y_0$) is simply the product of the probability densities of a Gaussian distribution evaluated for each $x_i$. For the multi-state fit of $y_1$, the model extends to a mixture of 1D Gaussians whereby $\hat{\mathcal{L}}$ is computed as a linear combination of each $K$ Gaussian components, corresponding to each state $\phi_j \in \{\phi_1, \dots, \phi_K\}$ with $\phi_j = (\mu_j, \sigma_j)$ weighted by a mixing coefficient ($\pi_j$) (*Bishop, 2006*).

$$\mathcal{N}(x \mid \mu, \sigma) = \frac{1}{\sigma\sqrt{2\pi}}\exp\left(\frac{-(x-\mu)^2}{\sigma^2}\right) \tag{6}$$

$$\hat{\mathcal{L}} = \prod_{i=1}^{N}\sum_{j=1}^{K}\pi_j * \mathcal{N}(x_i \mid \mu_j, \sigma_j) \tag{7}$$

To test the null hypothesis that $x$ is described by one state instead of two, BIC values are computed for $x$ with a fit of a single-state ($BIC_1$) and fit with two-states from divisive segmentation ($BIC_2$). If $BIC_2$ > $BIC_1$, $H_0$ is accepted and we believe $x$ is sufficiently described by a single-state. If the $BIC_2$ $\leq$ $BIC_1$, the $H_0$ is rejected and $x$ is split into two states. Assuming two-states are identified on the first iteration, the sequential process of CP detection and bi-partitioning with k-means clustering continues in a recursive fashion within data points belonging to each of the newly identified states (*Figure 1b*; *Pelleg and Moore, 2000*). Divisive segmentation continues to identify sub-clusters within each identified cluster until no cluster can be further partitioned. Overall, the recursive bi-partitioning and binary decision making in both CP detection and divisive clustering result in a reduced time complexity on the order of O($N$log($N$)).

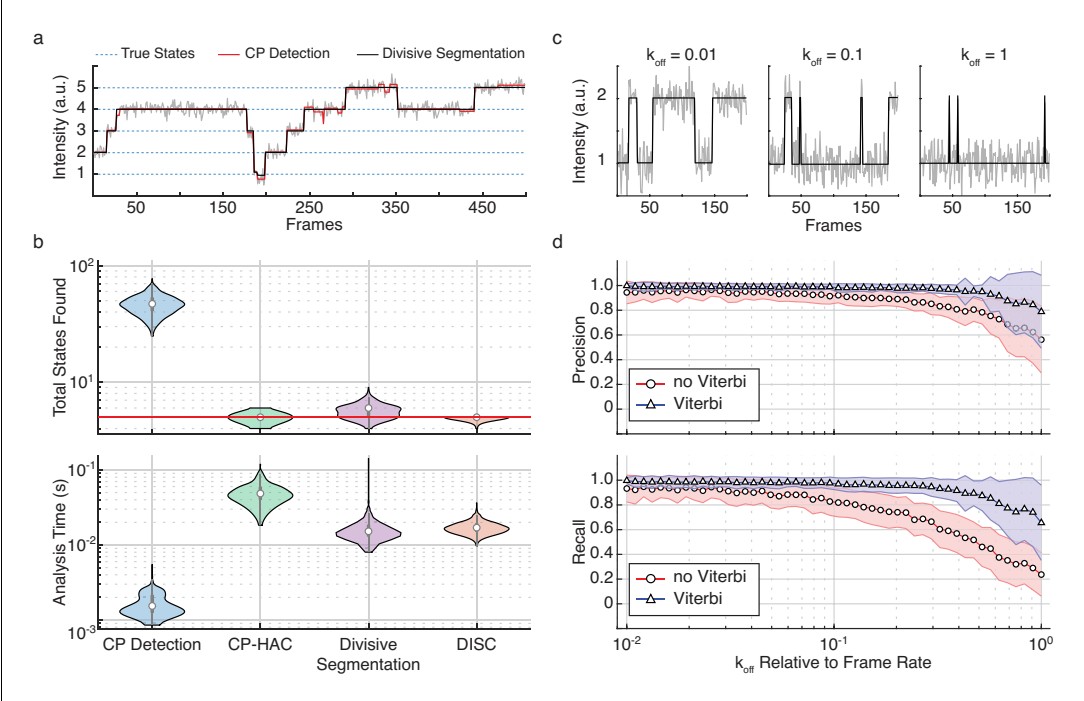

**Figure 2.** Refinement of divisive segmentation. (**a**) Example trajectory simulated with five states (blue) at a SNR = 5 overlaid with fits obtained from change-point detection (red) and divisive segmentation (black). (**b**) Violin plots showing the number of identified states (top) and analysis time (bottom) of each algorithm across 500 simulated trajectories featuring five true states (red line). (**c**) Example simulations of a two-state system with a $k_{on}$ = 0.02 frames$^{-1}$ and varying $k_{off}$. (**d**) Precision (top) and recall (bottom) values obtained with CP detection (no Viterbi) and Viterbi refinement obtained across 100 trajectories per $k_{off}$ (mean ± s.d.).

The online version of this article includes the following source data and figure supplement(s) for figure 2:

**Source data 1.** Simulated data of varying dynamics.

**Figure supplement 1.** The effect of state occupancy on DISC.

## Agglomerative clustering

Self-termination of divisive segmentation results in a series of estimated states and transitions within the trajectory. While this algorithm is exceptionally fast owing to its top-down greedy design, the reliance on local choices for state assignments rather than evaluation of the entire trajectory for an optimal decision can produce sub-optimal results. This error often surfaces as an over-sampling of the number of states and an under-sampling of the kinetic transitions. Over-fitting the number of states results from a downward dissemination of error from early splits: if bisecting a given cluster is suboptimal, two different parent clusters may each produce highly similar and redundant sub-clusters thereby overfitting the number of states. Fortunately, this over-estimate of the number of states can be corrected using bottom-up clustering. Therefore, the second phase of DISC uses HAC to compute the similarities between all identified states at a global level and assess the fit of the whole trajectory rather than segmented portions. Like CP-HAC schemes, an objective function is used to determine the overall fit and number of states in the trace by merging highly similar clusters whose separation may arise during divisive segmentation (*Figure 1b*). For a general application, we continue to use BIC for evaluating fit vs complexity. The similarity between neighboring states $\phi_i$ and $\phi_j$ is computed using Ward's minimum variance method (*Ward, 1963*), which considers the number of data points in each state (*n*) and the Euclidean distance between the means of the states,

$$d(\phi_i, \phi_j) = \sqrt{\frac{2n_{\phi i}n_{\phi j}}{n_{\phi i} + n_{\phi j}}} \left\| \mu_{\phi i} - \mu_{\phi j} \right\|_2 \tag{8}$$

The improvement of HAC on divisive segmentation for state detection is shown in *Figure 2*. Although divisive segmentation alone tends to slightly over-estimate the number of states, it provides a more reasonable estimate than CP detection alone (*Figure 2a*). The comparative performances in terms of speed and accuracy of these algorithms is further explored in *Figure 2b*. While CP detection alone (*Figure 2b*, blue) is very fast, it consistently yields a higher number of total states as compared to the ground truth. As CP-HAC frameworks must explore this large state space in its entirety, they can achieve higher accuracy than CP detection alone, but they are much slower algorithms (*Figure 2b*, green). In contrast, the use of top-down clustering in divisive segmentation dramatically reduces the total state-space for exploration, resulting in a faster algorithm than CP-HAC with much higher accuracy than CP detection alone (*Figure 2b*, purple). Finally, the sequential combination of divisive segmentation and HAC used in DISC lead to the highest state detection accuracy with minimal computational cost (*Figure 2b*, orange).

## Viterbi algorithm

Following state refinement with HAC, the trajectory is again described as a series of temporal transitions between identified intensity states. Although the overall states are well estimated at this point, fast transitions are often missed during CP analysis of single-molecule trajectories (*Hadzic et al., 2018*). To ensure events are accurately detected, the final phase of DISC applies the Viterbi algorithm (*Viterbi, 1967*).

The goal of the Viterbi algorithm is to identify the most probable sequence of hidden states through a series of observations. In our scenario, we have $K$ total states and $N$ total observations in our trajectory . In a naïve manner, determining the most probable sequence of hidden states $y$ could be accomplished by evaluating the likelihood of every possible hidden state sequence and choosing the most probable. However, as there are $K^N$ possible paths though the trajectory, this quickly becomes computationally intractable. A solution to this problem is the Viterbi algorithm, which makes use of dynamics programming to store only the most optimal state sequencing leading up to a given time point. In general, the Viterbi algorithm uses the observation that that the most probable state sequence leading up to data point n can be deduced by examining the most probable path leading up to the previous time point, *n-1*. Dynamic programming is used to keep track of all the optimal state sequences leading to all possible states for a given time point *n-1* which reduces the amount of required computations. Since there are there are $K$ states at time step *n-1*, the Viterbi algorithm stores $K$ possible state sequences leading up the previous time point *n-1*. At time point *n*, there are now $K^2$ paths to consider, given $K$ possible paths leading out of $K$ states. By examining the optimal sequence up to time point *n* and considering the probability of state transition between time points *n-1* and *n*, the most optimal state sequence up to time point *n* can be constructed. The state assignment of the first data point in the sequence can be determined by a provided initial probability of observing each state. Therefore, the time complexity of idealization with Viterbi is quadratic in the number of states $K$ and linear with the number of observations $N$, $O(K^2N)$, which is dramatically lower than an exhaustive search.

Formally, the Viterbi algorithm is described with a $K \times N$ trellis for states $j \in K$ and observations $n \in N$ (*Figure 1a*). Each cell of trellis $v_n(j)$ represents the probability of being in state $j$ after seeing the first $n$ observations and passing through the most probable state sequence for the given model parameters, $\lambda$. The value $v_n(j)$ is computed by recursively taking the most probable path up to this cell by

$$v_n(j) = \max_{y_1,\dots y_{n-1}} p(y_1 \cdots y_{n-1}, x_1 \cdots x_n, y_n = j | \lambda) \tag{9}$$

where $\lambda$ is first order Markov process of $\lambda = (\pi, a, b)$. The primary components of a hidden Markov model include the initial probability of observing each state $\phi_j$ given by $\pi$ where $\sum_{j=1}^{K} \pi_j = 1$; a transition probability matrix $a$ of size $K \times K$ where each element $a_{ij}$ is the probability of moving from $\phi_i$ to $\phi_j$, each element $a_{ii}$ is the probability of staying in $\phi_i$ and $\sum_{j=1}^{K} a_{ij} = 1$; and an emission probability matrix $b$ of size $K \times N$ where each element $b_j(x_n)$ is the probability of an observation $x_n$ arising from $\phi_j$. The values of each component are computed for each trajectory using the fits obtained from

sequential steps of divisive segmentation and HAC. Using these parameters, we can compute the most probable path for arriving in $\phi_j$ at time points $n$ by the following recursion

$$v_n(j) = \max_{i \in N} \{ v_{n-1}(i) a_{ij} b_j(x_n) \} \tag{10}$$

$$\psi_j(n) = \operatorname*{argmax}_{i \in N} \{ v_{n-1}(i) a_{ij} b_j(x_n) \} \tag{11}$$

where $\psi_j(n)$ is a helper function to store the $n-1$ state index $i$ on the highest probability path.

Upon termination, the forward likelihood of the entire state sequence $y$ up to time point N+1 having been produced by the given observations and HMM parameters is

$$P(y|x,\lambda) = \max_{i \in N} \{ v_T(i) \} \tag{12}$$

Deducing the most probable hidden state sequence $y$ through observations $x$ can be accomplished in a backtracking step by

$$y_n = \psi_n(y_n + 1) \qquad N \geq n \geq 2 \tag{13}$$

To assess the improvement of the Viterbi algorithm for event detection, we simulated a two-state system with a constant $k_{on}$ and varying $k_{off}$ rate (*Figure 2c*). As shown previously (*Hadzic et al., 2018*), we found that results from CP detection alone were accurate for slower events, but often failed to identify faster transitions (*Figure 2d*). By refining the results of unsupervised clustering with the Viterbi algorithm, we found that event detection accuracy was significantly improved over CP detection and clustering alone across two-orders of magnitude of varying $k_{off}$ (*Figure 2d*). Notably, as the changes in rates also affect the change in state occupancy, the high accuracy values returned after Viterbi refinement further highlight the power of DISC for resolving short-lived and rare transitions (*Figure 2—figure supplement 1*). In general, this improvement was anticipated since we are not the first to apply the Viterbi algorithm to the problem of idealization using unsupervised clustering. Although commonly used in the application of HMMs for hard assignment of data-points into $K$ discrete states, the SKM algorithm has shown that the Viterbi algorithm can successfully decode a path sequence following state clustering using k-means as opposed to more rigorous HMM training procedures (*Juang and Rabiner, 1990*; *Qin, 2004*). Therefore, both SKM and DISC can yield the event detection power of standard HMM approaches without the need of rigorous model training. However, unlike SKM, DISC has the added benefit of identifying the states naturally without the need for any user supervision such as initial state specification. This makes DISC a powerful alternative as a computationally efficient unsupervised single-molecule analysis algorithm.

## Modular nature of DISC

While the parameters used above are valid for trajectories with Gaussian noise, we do not claim they are optimal for all experimental modalities. We have intentionally developed DISC as a flexible framework for adaptation to different types of data. Although we used the Student's t-test for CP detection in the presence of Gaussian noise, additional merit functions may be more appropriate in different situations (*Song and Yang, 2017*; *Watkins and Yang, 2005*; *Li and Yang, 2019*). The same holds true for the use of BIC for state selection as other objective function can be substituted as needed. For example, the harsh penalty for parameters in BIC can lead to underfitting in certain cases; therefore, less stringent Akaike information criterion (AIC) or Hannan-Quinn information criterion (HQC) may be more appropriate depending on the separation of states and noise (*Akaike, 1974*; *Hannan and Quinn, 1979*). It is important to note that while DISC performs idealization through unsupervised clustering, obtaining accurate results does require the user to determine the appropriate information criterion and CP detection methods as idealization results heavily depend on these variables. Critically, the central innovation of DISC is to take advantage of the best features of both top-down and bottom-up forms of cluster identification that leads to both fast and accurate state detection.

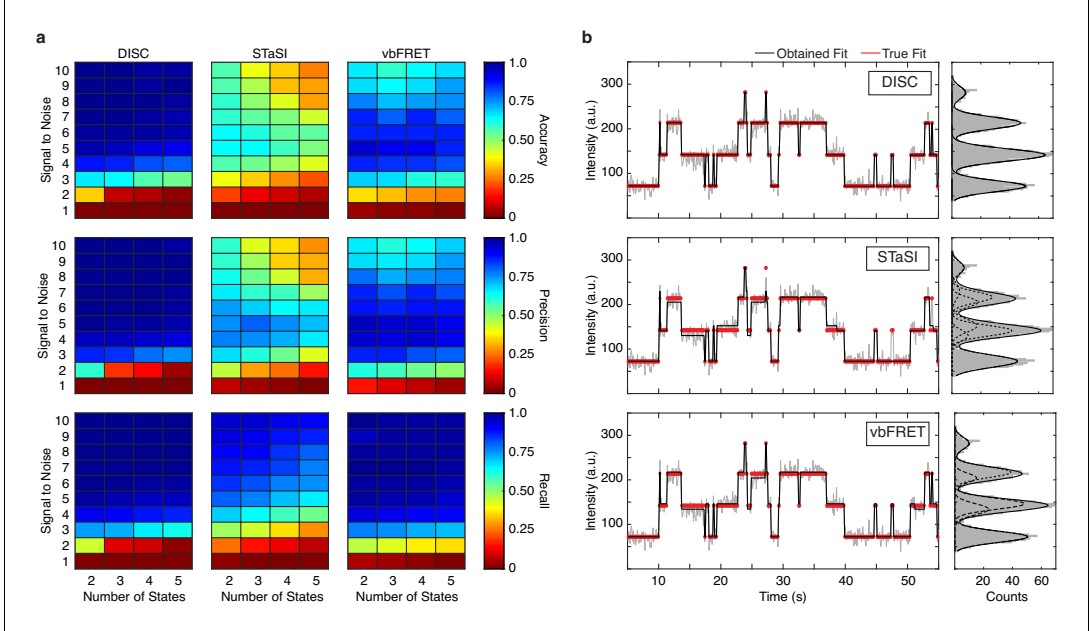

**Figure 3.** Standardizing performance. (**a**) Average accuracy (top), precision (middle) and recall values (bottom) computed for DISC, STaSI, and vbFRET across 100 trajectories at the specified signal to noise and number of states (Materials and methods). (**b**) Example simulated trajectory with four true states (red) fit and added Gaussian noise (grey) to SNR = 6 overlaid with fits (black) from DISC (top), STaSI (middle), or vbFRET (bottom).

The online version of this article includes the following source data and figure supplement(s) for figure 3:

**Source data 1.** Algorithm results of CNBD simulations.
**Figure supplement 1.** Characterization of fcAMP binding to monomeric CNBDs in ZMWs.
**Figure supplement 2.** Simulation with heterogenous fcAMP emission.
**Figure supplement 3.** Algorithm performance on simulations without heterogenous fcAMP emission.

# Results

## Validation of DISC on simulated data

We validate DISC using simulated single-molecule trajectories using kinetic parameters obtained from our recent studies exploring the regulatory mechanisms of cyclic nucleotide binding domains (CNBDs) from hyperpolarization-activated cyclic nucleotide gated ion channels (HCN) which regulate pacemaking in heart and brain cells (Materials and methods) (*Goldschen-Ohm et al., 2017*; *Goldschen-Ohm et al., 2016*). In these experiments, isolated CNBDs are tethered into ZMWs whereupon we monitor the binding and unbinding dynamics of fluorescent cyclic nucleotides (e.g. fcAMP) at physiological concentrations to uncover the elementary dynamics associated with channel gating. While ligand binding has been observed at the single-molecule level via both FRET and CoSMoS (co-localization), we adapt our simulations to the latter case so our dynamics are not limited in time by acceptor photobleaching. Notably, trajectories obtained with CoSMoS exhibit heterogeneous bound intensity values which vary with each binding event (*Figure 3—figure supplement 1*). While we are uncertain as to the exact source of this fluctuation, it is likely caused by shifts of the molecule in the heterogeneous excitation field of the ZMW or dye photodynamics (*Levene et al., 2003*; *Dempsey et al., 2009*). While the excitation field changes particularly sharply in ZMWs, TIRF and confocal microscopy also contain a heterogenous excitation field (*Moerner and Fromm, 2003*). Minor changes in apparent dye brightness due to dye conformational or photodynamics (such as in Protein-induced fluorescence enhancement, PIFE), shifts of dye orientation, or partial quenching via electron transfer are all commonly observed (*Stennett et al., 2015*). Thus, heterogeneous intensity values are a common and inconvenient feature in real life single-molecule fluorescence data.

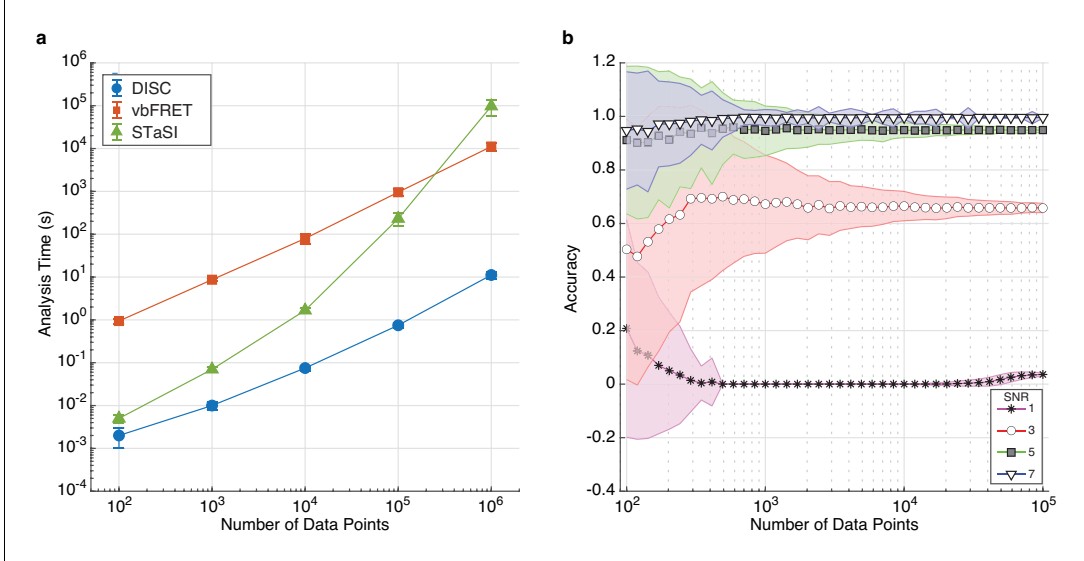

**Figure 4.** The effect of trajectory length on DISC performance. (a) Computational time (mean ± s.d.) of each algorithm for analyzing single trajectories of varying lengths. The test was performed with an Intel Xeon, 3.50 GHz processor running MATLAB 2017a. (b) Accuracy (mean ± s.d., N = 5000) of DISC for simulated trajectories of a two-state model with varying SNR and total number of data points.

The online version of this article includes the following source data and figure supplement(s) for figure 4:

**Source data 1.** Algorithm results across varying trajectory length.
**Figure supplement 1.** Algorithm performance on simulated smFRET data.

Including this additional noise source in our simulations yields a closer representation of experimentally obtained data.

In total, we simulated 4000 trajectories composed of 2000 data points each, totaling $8 \times 10^6$ data points. Each trajectory is 200 s in duration collected at frame rate of 10 Hz. We varied the complexity of the trajectory by simulating one to four independent CNBDs inside a given ZMW (two to five intensity states) and vary the signal to noise ratio (SNR) according to typical values from a ZMW experiment using Gaussian noise (Materials and methods) (*Goldschen-Ohm et al., 2017*; *Goldschen-Ohm et al., 2016*). We include the observed heterogenous bound intensities by randomly modulating each binding event according to our fit of the experimentally observed data (*Figure 3—figure supplement 2*). Given that each simulation features a different number of possible states, the total time spent within each state changes; therefore, these simulations also address the ability to capture states with unequal and even rare observation probabilities (see *Figure 3—source data 1*). We benchmark the results of DISC against commonly used HMM and CP-HAC methods: vbFRET and STaSI (*Bronson et al., 2009*; *Shuang et al., 2014*). These algorithms were chosen following the results of a recent comparative study that determined these to be the best performers among their class of analysis methods (*Hadzic et al., 2018*). In addition, DISC, STaSI and vbFRET all perform trajectory-by-trajectory idealization and are written entirely in MATLAB (MathWorks) which standardizes computational performance (Materials and methods).

Across all the simulations, DISC provides the highest average accuracy, precision and recall (*Figure 3a*, terms defined in Materials and methods). While no algorithm can idealize a trajectory in the presence of SNR = 1, DISC returns the lowest accuracy at SNR = 2. We suspect this result is due to the use of robust BIC for state detection; accuracy would likely be improved with less penalizing objective functions, such as AIC. While vbFRET performs the best at SNR = 2, the overall accuracy is still quite low: an average accuracy value for each number of simulated states is near chance. This low value demonstrates the inability of many algorithms to analyze data in presence of high noise and reinforces the common practice of discarding noisy data to create a more reliable dataset. For SNR > 3, which accounts for most of our experimentally obtained data (*Figure 3—figure supplement 1c*), DISC performs exceptionally well with highest average accuracy (0.91 ± 0.05) and is robust

against false positives (precision = 0.96 ± 0.04) and false negatives (recall = 0.93 ± 0.03) across all simulated conditions (*Figure 3a*). While vbFRET matches the recall of DISC in this SNR range (0.94 ± 0.05), the tendency to overfit the number of states at higher SNR lowers precision (0.80 ± 0.18) and overall accuracy (0.76 ± 0.19). We find STaSI returns the lowest overall accuracy (0.47 ± 0.17) likely do to an overfitting the number of states (precision = 0.57 ± 0.2) and a tendency to miss transitions (recall = 0.75 ± 0.10). Notably, DISC is the only method unaffected by inclusion of heterogeneous state intensities of fcAMP likely due to the use of a Gaussian derived BIC for state selection (*Figure 3—figure supplement 3*).

Critically, DISC not only returned high accuracy results, DISC was also much faster than the other methods. Idealization of all 4000 trajectories by DISC was completed in just over two minutes, whereas STaSI took over fifteen minutes and vbFRET took over twelve hours. To thoroughly explore the computational efficiency of DISC, we simulated data with increasing durations per trajectory at a constant SNR and number of states. Remarkably, we find DISC is 400-fold to 1,200-fold faster than vbFRET and 2-fold to 8,700-fold faster than STaSI due to STaSI's quadratic time dependence (*Figure 4a*; *Shuang et al., 2014*). For example, a trajectory of $10^6$ data points can be analyzed by DISC in 10 s compared to 3 hr for vbFRET and 27 hr for STaSI. Thus, DISC can handle the analysis of long trajectories, unlike CP-HAC methods. While fluorescence measurements from a single fluorophore at room-temperature rarely contain this many data points, large trajectory lengths are common in non-fluorescence experiments or fluorescence experiments with replenishing fluorescent labels such as in single-molecule genome sequencing and studies of catalysts via fluorogenic reactions (*Eid et al., 2009*; *English et al., 2006*; *Sambur et al., 2016*). To evaluate performance on more typical data, we compared the results of each algorithm on simulated smFRET trajectories that are limited in duration by acceptor photobleaching (*Figure 4—figure supplement 1*, Materials and methods). For simulations featuring two or three states FRET, we find DISC 5.5-fold faster than STaSI and 235-fold faster than vbFRET while maintaining the highest accuracy. This result validates the use of DISC for the analysis of large volumes of shorter fluorescence trajectories, especially compared to HMM approaches. This feature is particularly important as advances in hardware such as CMOS cameras and lab-on-chip methods generate larger smFRET data sets (*Juette et al., 2016*).

Finally, we evaluate the effect of trajectory duration on DISC accuracy. As expected, we find that the accuracy increases with increasing number of data points per trajectory. This result also indicates the minimum number of data points needed for an accurate idealization for a given SNR (*Figure 4b*). Overall, the results of our simulations suggest DISC is more or comparably accurate and critically, is substantially faster than standard idealization approaches, making it an enabling technology for analysis of high-throughput single-molecule experiments.

## The binding of cAMP to HCN CNBDs is non-cooperative

To verify performance of DISC in an experimental configuration with high volumes of experimental data, we analyzed a large single-molecule data set obtained from ZMWs that explore HCN dynamics (*Figure 5a*). Previous macroscopic studies of HCN channel gating have revealed that ligand binding to CNBDs exhibits both positive and negative cooperativity depending on the ligation state and the membrane potential (*Kusch et al., 2012*; *Thon et al., 2015*). Cyclic AMP regulates cardiac pacemaking via HCN channels and, therefore, this unusual allostery has significant physiological implications. However, as this allosteric analysis was based on global fits of ensemble binding data, the reliability of model parameters remains an open question (*Hines et al., 2015*). Evaluating cooperativity is an experiment well-suited for single-molecule investigation given the ability to observe the total time a molecule spends in each liganded state and directly extract state transition probabilities. Therefore, to directly assess the cooperativity between HCN2 CNBDs upon ligand binding, we use single-molecule fluorescence and monitor the binding of individual fcAMP molecules to our previously described tetramerized CNBDs inside ZMWs. (*Figure 5a*; *Goldschen-Ohm et al., 2016*).

Our initial dataset included 13,670 ZMWs each monitored for 800 s at a sampling rate of 10 Hz (Materials and methods). All trajectories were obtained in the presence of 1 µM fcAMP which is near the ligand dissociation constant for individual CNBDs (*Goldschen-Ohm et al., 2016*). As shown with other high-throughput collection platforms, an essential part of analysis at this scale is the application of stringent criteria to select traces that yield meaningful information about the system (*Chen et al., 2014*; *Juette et al., 2016*). Therefore, we first analyzed all trajectories with DISC to

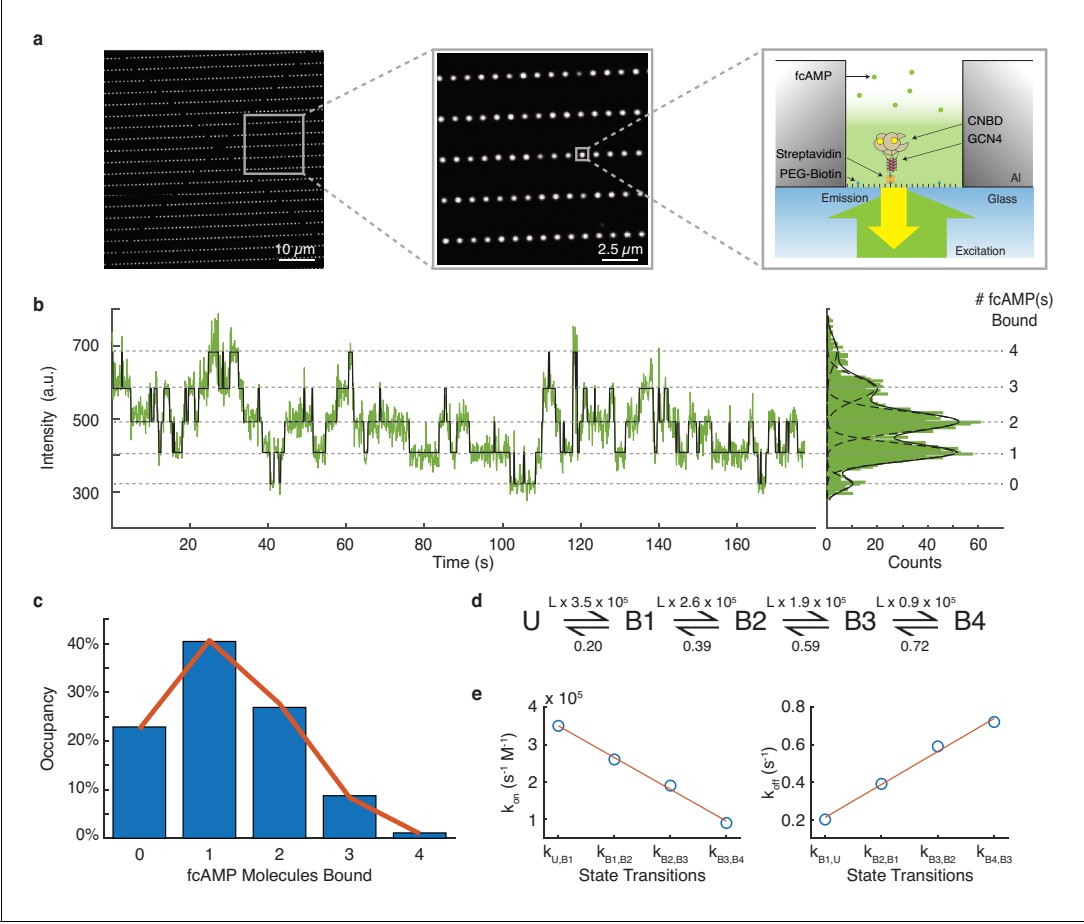

**Figure 5.** DISC analysis of HCN CNBDs. (**a**) Representative ZMW arrays for observing fcAMP binding to tethered tetrameric CNBD. (**b**) Representative time series of 1 μM fcAMP binding to tetrameric CNBD fit with DISC with up to four fcAMP molecules binding simultaneously. (**c**) Observed distribution of fcAMP occupancy fit with a binomial distribution (orange). (**d**) Sequential model of four binding steps and one unbound state with globally optimized rate constants. The rate constants are given as $s^{-1}$ or $s^{-1}M^{-1}$ where L is the ligand concentration in M. (**e**) Linear regression of rate constants $k_{on}$ ($m = -8.5 \times 10^{4} \ s^{-1}M^{-1}$, $b = 4.35 \times 10^{5} \ s^{-1}M^{-1}$, $R^{2} = 0.99$) and $k_{off}$ ($m = 0.18 \ s^{-1}$, $b = 0.035 \ s^{-1}$, $R^{2} = 0.99$) for each sequential state.

The online version of this article includes the following source data and figure supplement(s) for figure 5:

**Source data 1.** Plotted data and fits of tetrameric CNBD dynamics.
**Figure supplement 1.** Tetrameric CNBD analysis by DISC and STaSI.
**Figure supplement 2.** Non-specific fcAMP binding in ZMWs.
**Figure supplement 3.** Asynchronous decay of tetrameric CNBD activity over excitation time.
**Figure supplement 4.** Example trajectories of 1 μM fcAMP binding to tetrameric CNBDs in ZMWs.
**Figure supplement 5.** Model comparison of HCN CNBDs.

find reliable data prior to trace selection. DISC successfully processed this entire data set within 20 min using a standard MacBook Air (1.6 GHz Intel Core i5). The same analysis completed with STaSI yielded unphysical results in 4 hr (*Figure 5—figure supplement 1*). We estimated analysis with vbFRET would take weeks to complete and was therefore not performed. While correcting for non-specific binding is often a necessity in CoSMoS experiments, we find the passivated surfaces within the ZMWs greatly reduce non-specific absorption of fcAMP to either the metallic or glass surfaces, thus minimizing this concern (*Figure 5—figure supplement 2*; *Smith et al., 2019*; *Eid et al., 2009*; *Foquet et al., 2008*). Using the idealized fits obtained from DISC, we screened our data to select reliable trajectories for our analysis. Standard cut-offs in state separation, the total number of observed states, and a filter for kinetic activity ensured that each trajectory arose from a ZMW featuring a singly occupied and functional tetrameric CNBD. Notably, we noticed an asynchronous

decay of protein activity over excitation time that may be caused by singlet-oxygen formation and subsequent inhibition of cyclic nucleotide binding to CNBDs (*Figure 5—figure supplement 3*; *Idikuda et al., 2018*). In conclusion, we retained 293 molecules totaling $1.2 \times 10^5$ s of combined protein activity across 53,474 events.

To determine if the binding of cAMP to CNBDs is cooperative, we first calculated the total time each molecule spends in each of the liganded states (0 to 4 fcAMPs) using the state assignments from the idealized fits (*Figure 5b*, *Figure 5—figure supplement 4*). The resulting distribution of state occupancies is fit with a binomial distribution to evaluate the independence of each CNBD (*Figure 5c*, Materials and methods). Our binomial fit matches the distribution well and returns the probability of occupancy at 1 μM fcAMP for a single CNBD in the tetramer as 31% which is similar to our previous monomeric CNBDs studies (*Goldschen-Ohm et al., 2016*), suggesting a lack of cooperativity between the CNBDs. Notably, our measured state-occupancy distribution is strikingly different that than the unusual cooperativity modeled from either activated or non-activated channels (*Figure 5—figure supplement 5*; *Kusch et al., 2012*; *Thon et al., 2015*). We further explored the underlying dynamics of our data using the idealized single-molecule transitions obtained from DISC. Using QuB, we built a simple HMM of sequential ligand binding across four binding sites that was globally optimized across each molecule's idealized state trajecotry (*Nicolai and Sachs, 2013*; *Qin et al., 2000*; *Figure 5d*, Materials and methods). As expected for non-cooperative processes, the optimized $k_{on}$ and $k_{off}$ rates for each state transition exhibit a strong linear relationship (*Figure 5e*). Combined, these results strongly suggest that CNBD units act independently during ligand binding. We postulate that the macroscopically observed cooperativity is either an artifact of model fitting or that it requires the presence of the transmembrane domains of the HCN channel and is not an intrinsic property of the CNBDs.

## Discussion

We developed a new algorithm for rapid and accurate unsupervised idealization of single-molecule trajectories. Our approach combines unsupervised statistical learning of discrete states with the event detection power of the Viterbi algorithm to quickly identify both significant states and transitions in a model-independent manner. Software implementing the DISC algorithm that includes a graphical user interface is available at https://github.com/ChandaLab/DISC (*Chanda et al., 2019*; copy archived at https://github.com/elifesciences-publications/DISC).

Like CP-HAC methods, DISC is not a fully probabilistic approach. While fully probabilistic HMM training approaches are beneficial for providing unbiased estimates of the parameter distributions, their high accuracy comes at the cost of significantly increased computational time. This cost is especially apparent for the recently developed infinite HMM approaches that aim to learn the true number of states from a potentially infinite number of possibilities. However, accomplishing this task costs hundreds of iterations per trace to provide a reproducible fit with some approaches taking days to analyze single trajectories (*Hines et al., 2015*; *Sgouralis and Pressé, 2017*; *Sgouralis et al., 2018*). Thus, while exhaustive search algorithms may be desirable in other contexts, they are clearly not suited for large datasets associated with high-throughput experiments. In contrast, by simulating data closely resembling the binding of fcAMP to pacemaker channels, we find that DISC surpasses the accuracy of common CP-HAC and HMM algorithms with a dramatic improvement in computational speed. Therefore, DISC satisfies the need for accuracy and speed in high-throughput analysis. In this regard, DISC is like the SKM algorithm for estimating the parameters of an HMM without direct HMM training. However, unlike SKM which relies on user-specified states, DISC uses unsupervised statistics to learn the states. Therefore, DISC offers the idealization power of SKM with the state-learning capabilities of CP-HAC.

We used DISC to analyze a large dataset obtained from ZMWs to evaluate cooperativity between CNBDs from pacemaker ion channels upon ligand binding. The rapid and robust idealization provided by DISC enabled stringent trace selection to ensure only reliable trajectories were analyzed. These data show that, in contrast to model inferences from ensemble measurements of HCN2 channels, CNBD tetramers do not exhibit cooperative ligand binding. This result suggests that allosteric interactions between binding sites may be coordinated by the channel's transmembrane domains.

Although we have demonstrated the use of DISC on single-molecule fluorescence data, the framework can be easily extended to other data paradigms due to its modular nature. For example,

the use of BIC for state determination or the Student's t-test for change-point analysis could be interchanged with other information theoretic approaches or merit functions where appropriate. To allow for easy comparison of a given data set, the provided software and graphical user interface (GUI) allows the user to select from several options the desired parameters such as choice of information criteria. This flexibility makes DISC suitable for a wide array of experimental data provided it can be described as a series of transitions between discrete states, including, for example, single-channel current recordings, force spectroscopy and smFRET. However, there is no inherent knowledge within DISC to consider various sources of experimental noise, such as photo-blinking or baseline drift; therefore, correcting for these noise sources prior to DISC analysis will likely improve idealization accuracy.

Finally, our results show that DISC provides a dramatic improvement in computational speed over current state-of-the-art approaches while either improving or maintaining high accuracy for both state determination and event detection. This increase in speed is directly applicable to analyzing the growing datasets obtained in single-molecule fluorescence paradigms to adequately sample population dynamics. For example, the use of sCMOS camera enables smFRET measurements of tRNA conformational changes during protein translations across thousands of molecules simultaneously with millisecond resolution (*Juette et al., 2016*). Additionally, magnetic tweezers have enabled week-long mechanical measurements of single-protein folding and unfolding, shifting observable dynamics to pathological time-scales and allowing the detection of rare events (*Popa et al., 2016*). Thus, highly computationally efficient and robust algorithms such as DISC may be well suited for analysis of a wide variety of single molecule datasets beyond the standard smFRET data.

# Materials and methods

## Key resources table

| Reagent type (species) or resource | Designation | Source or reference | Identifiers | Additional information |
|---|---|---|---|---|
| Biological sample | GCN4pLI-HCN2 tetramer | DOI: 10.7554/eLife.20797 | | |
| Chemical compound, drug | Protocatechuate 3,4-Dioxygenase | Millipore-Sigma | cas no. 9029-47-4 | from *Pseudomonas* sp. |
| Chemical compound, drug | 3,4-Dihydroxybenzoic acid | Millipore-Sigma | cas no. 99-50-3 | Protocatechuic Acid |
| Chemical compound, drug | (±)-6-Hydroxy-2,5,7,8-tetramethylchromane-2-carboxylic acid | Millipore-Sigma | cas no. 53188-07-1 | Trolox |
| Chemical compound, drug | 8-[DY-547]-AET-cAMP | BIOLOG | Cat. No.: D 109 | DOI: 10.1016/j.neuron.2010.05.022 |
| Software, algorithm | MATLAB | MathWorks | RID:SCR_001622 | |
| Software, algorithm | QuB | DOI: 10.1142/1793048013300053 | | https://qub.mandelics.com/ |
| Software, algorithm | DISC | This work | | https://github.com/ChandaLab/DISC |
| Software, algorithm | vbFRET | DOI: 10.1016/j.bpj.2009.09.031 | | http://vbfret.sourceforge.net/ |
| Software, algorithm | STaSI | DOI: 10.1021/jz501435p | | https://github.com/LandesLab/STaSI |
| Other | Zero-Mode Waveguide | Pacific Biosciences | Non-Commercial Zero-Mode Waveguides (Astro Chips) | |

## Single-molecule simulations

Single-molecule trajectories were simulated as a Markov process of transitions between discrete states. All simulations were performed with a frame rate of 10 Hz and featured variable total durations, SNR, and number of states. The primary kinetic scheme used was adapted from our recent studies of fcAMP binding to isolated monomeric CNBDs (*Goldschen-Ohm et al., 2016*). This model is a four-state scheme where both the unbound (U) and bound states (B) exhibit conformational changes (U' ⇔ U ⇔ B ⇔ B'), yet exhibit only two different observable states (ie, U'/U are indistinguishable via fluorescence intensity, as are B/B'). fcAMP binding occurs between U and B. The rate constants (s$^{-1}$ or M$^{-1}$ s$^{-1}$) are: $k_{U'U}$ = 0.15; $k_{U,U'}$=0.04; $k_{U,B}$ = 2.3x10$^{-6}$ * [fcAMP]; $k_{B,U}$ = 0.95; $k_{B,B'}$=0.51; $k_{B',B}$ = 0.31 at 1 µM fcAMP. To mimic the tetrameric nature of HCN channels with no cooperativity, we extrapolated up to four bound states by summing independent CNBD trajectories prior to the addition of noise. To include realistic SNR, state-intensities, and heterogeneity distribution of bound intensities, we analyzed the direct fcAMP excitation and emission trajectories following acceptor photobleaching from the monomeric CNBD dataset used in our previous work (*Figure 3—figure supplement 1*; *Goldschen-Ohm et al., 2016*). This dataset consisted of 861 single molecules for a combined acquisition time of 44,090 s (4775 total binding events). All trajectories had a SNR >2 and all events persisted for longer than two frames, which resulted in an imbalance in the bound and unbound events. For each simulated trajectory, state intensities were each drawn from log normal distributions fit to monomeric CNBD single-molecule data, with average intensities between subsequent states being uniform. Gaussian noise was applied to trajectories at specified SNR. To quantitate the heterogeneous intensities from fcAMP binding, the mean of individual bound event intensities were taken for each identified event, so long as the event was >2 frames in duration. Heterogeneity was computed as the absolute percent difference for each event vs the mean bond intensity for the given trajectory by:

$$Percent\ Heterogenity = \left| \frac{>_{event} - >_{bound}}{>_{bound}} \right| \times 100\% \tag{14}$$

The heterogeneity of unbound events was minimal and was therefore not included in the simulations. For each simulated event, heterogenous bound intensity emissions were each drawn from an exponential fit monomeric CNBD single-molecule data. Gaussian noise was added to trajectories as specified.

Simulated smFRET data were downloaded from the kinSoftChallenge on June 11$^{th}$, 2019 (https://sites.google.com/view/kinsoftchallenge/home). Data used came from the provided training data sets titled: 'Level 1' and 'Level 2' with folder names 'sim_190212_194543_level1' and 'sim_190212_202530_level2'.

## Algorithm performance

DISC, STaSI, and vbFRET are all written entirely in MATLAB (MathWorks). Each algorithm was used outside of their graphical user interfaces (GUIs) to more accurately compare the computational time of native functions within each algorithm. User parameters in DISC include: the confidence interval of CP detection and the objective function for clustering. Unless otherwise stated, a 95% confidence interval was applied for CP detection and BIC was used for all clustering. For analysis with STaSI and vbFRET, we used the recommended default values set by their authors (*Bronson et al., 2009*; *Shuang et al., 2014*). For STaSI, this means a 99.8% confidence interval of CP detection. In vbFRET, users must provide the number of states and fitting attempts per trace (left at the default value of 10). To circumvent providing the number of states, we modified the provided *vbFRET_no_gui.m* script to perform analysis outside of the vbFRET GUI. The modified script begins by fitting the trace to one state and increases the number of states until two more beyond the number of states with the maximum evidence to ensure the maximum fit has been obtained. As no changes were made to native vbFRET functions, implementing this script has no effect on vbFRET's accuracy. We expect changing parameters in both STaSI and vbFRET may lead to different results; however, it was not our goal to optimize the use of these algorithms. Also, as a thorough investigation into the performance of STaSI and vbFRET has been conducted elsewhere, we did not investigate why these algorithms presented lower performance than DISC (*Hadzic et al., 2018*).

All quantifications of computational time were performed using the tic and toc functions in MAT-LAB. For idealization accuracy, each event returned by a given algorithm is classified as a True Positive (TP), False positive (FP), or False Negative (FN). We define a TP as being in the correct state (±10% the correct intensity level's standard deviation) and correct event duration (±1 frame) for a given simulated event. FPs are either added events or correct events in the wrong state. FNs are missed events. For each trajectory, we computed accuracy, precision, and recall as:

$$Accuracy = \frac{TP}{(TP + FP + FN)} \tag{15}$$

$$Precision = \frac{TP}{(TP + FP)} \tag{16}$$

$$Recall = \frac{TP}{(TP + FN)} \tag{17}$$

Accuracy represents the overall performance, whereas precision and recall highlight the false positive error rate (overfitting the data) and false negative rate (underfitting the data), respectively.

## Single-molecule fluorescence microscopy in ZMWs

The expression, purification, biotinylation, and fluorescence labeling of tetrameric CNBDs were performed as previously described (*Goldschen-Ohm et al., 2016*). Non-commercial arrays of ZMWs were purchased from Pacific Biosciences. These waveguides featured a polyphosphonate passivation layer on the aluminum walls and a biotinylated polyethylene glycol (PEG) layer on the glass surface to reduce non-specific binding (*Foquet et al., 2008*; *Eid et al., 2009*). The PEG-Biotin surface was incubated with 0.05 mg/mL streptavidin (Prospec, cat # PRO-791) for 5 min in a buffer containing: 40 mM HEPES, 600 mM NaCl, 20% glycerol, 2 mM TCEP, 0.1% LDAO (Sigma, cat no. 40236), 2 mg/mL bovine serum albumin (BSA), 1 mM Trolox (Sigma, cas no. 53188-07-1), 2.5 mM protocatechuic acid (Sigma, cas no. 99-50-3) (PCA), pH 7.5 (Buffer A). After incubation, the ZMW chip was thoroughly rinsed with Buffer A to remove unbound streptavidin. Next, biotinylated tetrameric-CNBDs were diluted in Buffer A with the addition of the PCA/PCD oxygen scavenging system by adding 250 nM of protocatechuate 3,4-dioxygenase (PCD) from Pseudomonas sp. (Sigma, cas no. 9029-47-4) to between 100 pM and 2 nM for surface immobilization in ZMWs (Buffer B) (*Aitken et al., 2008*). This resulted in ≈100 occupied ZMWs out of the total ≈1000 ZMWs per field of view identified by fluorescence bleach steps of DY-650 that labels each of the four CNBDs. Fluorescently labeled cAMP (fcAMP; 8-(2-DY-547]-aminoethylthio) adenosine-3',5'-cylic monophosphate) (BioLog, cat # D 109) was added at 1 µM for all single-molecule experiments in Buffer B.

ZMW chips were placed on top of an inverted microscope (Olympus IX-71, 100X, NA 1.49) and imaged under 532 nm (60 W/ cm$^2$) or 640 nm (25 W/ cm$^2$) (Coherent) as described previously (*Goldschen-Ohm et al., 2017*; *Goldschen-Ohm et al., 2016*). The only notable difference is that unlike previous experiments we did not use FRET to monitor binding in order to obtain data for extended periods (*Goldschen-Ohm et al., 2016*). We excited DY-650 with 640 nm to identify ZMWs featuring DY-650-labeled tetrameric-CNBDs. Next, fcAMP was continuously imaged with 532 nm for 8000 frames at 10 Hz to monitor binding activity. All emission spectra were split with a 650 nm long pass dichroic (Semrock Brightline FF650) and bandpass filtered using pairs of edge filters (532–623.8 nm, 632.9–945 nm; Semrock Cy3/Cy5-A-OMF) and imaged onto two separate EMCCDs (Andor iXon Ultra X-9899) using Metamorph software (Molecular Devices). All data were collected using ZMWs of 150–200 nm diameter which are large enough to accommodate tetrameric CNBD complex (each monomeric CNBD is 4 × 6×20 nm) (*Goldschen-Ohm et al., 2016*).

## Single-molecule ligand binding image analysis

All analysis was performed using custom software written in MATLAB (Mathworks) or ImageJ. Single-molecule trajectories of each ZMW were extracted from tiff stacks saved by Metamorph software using MATLAB. Locations of ZMWs were obtained using a threshold mask of the brightfield image of the whole ZMW array. ZMW locations were refined with a 2D Gaussian fit to the local intensity height map. The time-dependent fluorescence at each ZMW was obtained by projecting the average

image intensity in a 5-pixel diameter circle onto the ZMW location throughout each image in the stack.

## Trace selection and analysis of binding activity

A total of 13,670 individual ZMWs ($1.1 \times 10^8$ data points) were processed from the single-molecule tetrameric CNBD experiments. Each trajectory was idealized with DISC using a 95% confidence interval for CP detection and BIC for state selection (*Figure 5—figure supplement 1*). To reflect the ability to cleanly resolve the individual occupation states, we computed the separation of each sequential state vs the noise within a state by:

$$State\ Separation = \frac{1}{K-1}\sum_{i=2}^{K}\frac{(\mu_i - \mu_{i-1})}{\sigma_{i-1}} \tag{18}$$

where K is the total number of states, $\mu$ is the mean intensity value of a state, and $\sigma$ is the standard deviation of the data points belonging to a state. This ensures that states are separated well enough to resolve, as would be expected for sequential ligand binding. Traces featuring 4 to 6 identified states with state separation $\geq 3$ were retained for further analysis.

To ensure a given trajectory contained a functional tetrameric CNBD, we kept traces that spent less than 50% of the time in the unbound state, resulting in a total of 480 trajectories for visual inspection. The observed asynchronous decay of protein activity was corrected using the CP detection method to identify the most likely point in a given trajectory where protein behavior dramatically changed. This was accomplished using MATLAB's findchangepoint function using the change in standard deviation as the statistic. Data points following the identified CP location were discarded from the analysis to include only the frames of consistent fcAMP binding to presumably functional proteins (*Figure 5—figure supplement 3*).

In conclusion, a total of 293 molecules totaling $1.2 \times 10^5$ s ($\approx 34.5$ hr) of combined protein activity across 53,474 events was included for the final analysis. Each trajectory exhibited four or five conformational states (3 to 4 fcAMPs bound). Binomial fitting of the total time spent in each state was performed using MATLAB's mle function. HMM modeling of single-molecule binding events was performed with QuB (*Nicolai and Sachs, 2013*; *Qin et al., 2000*). Idealized trajectories from DISC were exported to QuB with the first and last events removed. A sequential model of 0 to 4 ligand binding sites was globally optimized to simultaneously describe the idealized binding trajectories for all molecules.

## Acknowledgements

We thank Dr. Mike Sanguinetti for the wild-type HCN2 plasmid and Dr. Vadim A Klenchin for the purification of the tetrameric-CNBD. We also thank Owen Rafferty for his assistance in the development of the GUI for running the DISC algorithm.

## Additional information

### Competing interests

Baron Chanda: Reviewing editor, *eLife*. The other authors declare that no competing interests exist.

### Funding

| Funder | Grant reference number | Author |
| --- | --- | --- |
| National Institute of Neurological Disorders and Stroke | NS-101723 | Baron Chanda |
| National Institute of Neurological Disorders and Stroke | NS-081320 | Baron Chanda |
| National Institute of Neurological Disorders and Stroke | NS-081293 | Baron Chanda |

| National Institute of General Medical Sciences | GM007507 | David S White |
| National Institute of General Medical Sciences | GM127957 | Randall H Goldsmith |
| National Science Foundation | CHE-1856518 | Randall H Goldsmith |

The funders had no role in study design, data collection and interpretation, or the decision to submit the work for publication.

## Author contributions

David S White, Conceptualization, Resources, Data curation, Software, Formal analysis, Investigation, Methodology; Marcel P Goldschen-Ohm, Conceptualization, Software, Formal analysis, Investigation, Methodology; Randall H Goldsmith, Conceptualization, Supervision, Investigation, Methodology, Project administration; Baron Chanda, Conceptualization, Supervision, Funding acquisition, Investigation, Methodology, Project administration

## Author ORCIDs

David S White (iD) https://orcid.org/0000-0003-0164-0125
Marcel P Goldschen-Ohm (iD) https://orcid.org/0000-0003-1466-9808
Randall H Goldsmith (iD) https://orcid.org/0000-0001-9083-8592
Baron Chanda (iD) https://orcid.org/0000-0003-4954-7034

## Decision letter and Author response

Decision letter https://doi.org/10.7554/eLife.53357.sa1
Author response https://doi.org/10.7554/eLife.53357.sa2

# Additional files

## Supplementary files

• Transparent reporting form

## Data availability

Simulated and raw data in addition to analysis scripts are available at https://doi.org/10.5281/zenodo.3727917.

The following dataset was generated:

| Author(s) | Year | Dataset title | Dataset URL | Database and Identifier |
|---|---|---|---|---|
| White D S, Goldschen-Ohm MP, Goldsmith R, Chanda B | 2020 | Data Associated with Top-Down Machine Learning for High-Throughput Single-Molecule Analysis | http://dx.doi.org/10.5281/zenodo.3727917 | Zenodo, 10.5281/zenodo.3727917 |

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
