## [Decision Letter]

**Acceptance summary:**

This manuscript by White et al. describes a new statistical method for event detection in single-molecule biophysics. The performance of the new algorithm and workflow is higher than that of other well-established methods of single-molecule statistical analysis when probed against the same benchmarks. This new methodology combines a new top-down segmentation approach with other well-established event detection methods to yield a procedure that should accelerate and facilitate parameter extraction from single-molecule experiments.

**Decision letter after peer review:**

Thank you for submitting your article "High-Throughput Single-Molecule Analysis via Divisive Segmentation and Clustering" for consideration by *eLife*. Your article has been reviewed by three peer reviewers, including Leon D Islas as the Reviewing Editor and Reviewer #1, and the evaluation has been overseen by Olga Boudker as the Senior Editor. The following individual involved in review of your submission has agreed to reveal their identity: Fred Sigworth (Reviewer #2).

The reviewers have discussed the reviews with one another and the Reviewing Editor has drafted this decision to help you prepare a revised submission.

Summary:

The authors present us with a manuscript describing a new method to carry out idealization and parameter estimation of single-molecule data. The method merges several good-performing algorithms to segment the data and find transitions between states and finally discover the most probable number of states giving rise to the data. Importantly, they compare the performance of their DISC algorithm to other two well established methods and discover that DISC is faster and perhaps more accurate in certain experimental conditions.

The reviewers agree the DISC method is an important contribution to the arsenal of single-molecule trajectories analysis methods and the authors have provided a good characterization of its performance. The paper has merits and the method is an important contribution to the field, however, there are a number of important comments that were raised that need to be dealt with by the authors.

Essential revisions:

The paper needs to be rewritten in a manner that enough formal description of the method is given to allow the target readership (SM experimentalists, presumably) to understand how it works and what exactly it does. As it is, the paper reads more like an "advertisement" of an algorithm described in detail elsewhere.

1) The new Divisive Segmentation (DS) algorithm is described in general terms in the text but a formal description is not provided. A reasonable description of the CP analysis portion of DS is provided in Supplemental Note 2 (though a suitable pointer to this from the main text is not given). In the text you then note that you employ a modified k-means algorithm. The reader asks, what is the modification? The citations are unhelpful. Finally, the HAC part is also fairly well described in Supplemental Note 3. But the heart of the DS algorithm does not receive a similarly formal description. A similar problem arises with the mention of the Viterbi algorithm. We are told that you "use" it, but the only references are to the ion-channel and speech-recognition literature. Beside not explaining briefly what it does (for the naive reader's benefit), you do not explain what emission probability densities you use, nor how you set transition probabilities.

2) A question that arises regarding assumptions and parameters: how well does the DS/HSC perform for traces with rarely-visited states? Your selection of total fraction bound > 0.5 (subsection “Trace selection and analysis of binding activity”) suggests that your approach might be best for situations where the various states are roughly equi-probable. An important point to check is the performance of DISC as a function of the kinetics of the underlying process. How well does DISC estimates the number of states for fast kinetics, with a small number of data points in each state (short dwell-times)? One of the goals of single-molecule analysis is the determination not only of the most probable number of states, but also the dwell-times in each state and the transition rates between them. Was an effort made to include dwell-time estimation in DISC?

3) The authors need to be careful using the term high throughput as this means different things to different scientific fields. For instance, high throughput drug discovery often means looking at millions of compounds. Certainly 4000 molecules is relatively high throughput compared to a handful, but even the earliest single-molecule fluorescence papers included a hundred traces, so does a 40x increase justify the semantic transition to high throughput?

4) As authors show in the assessment of simulated traces in Figure 3, none of the methods compared work in a meaningful way when looking at SNR less than or equal to 3. The data simulated in the manuscript appears to be >10:1 according to a first approximation. So none of the methods assessed does a meaningful job revealing the 'true' number of states even when the data appear to be very good.

5) In the Introduction and Discussion sections, the authors downplay the general usefulness of existing Markov modeling methods. This is unfortunately misleading. The value of Markov modeling for time-series data analysis is proven in the extensive literature across a large number of fields using a variety of experimental approaches. While recent advancements in data throughput (e.g., sCMOS cameras) have put pressure on these algorithms to be efficient, this is a relatively minor problem that can be overcome by better implementations. Related to the above, Supplementary Note 1 includes a great deal of important introductory information that does a better job of placing DISC into the context of existing methods. This information belongs in the main text to substantively review the subject matter.

6) Given that the authors know (or should reasonably expect) that their experimental system has exactly five states, how does their "model free" approach compare to a traditional, model-based approach with a reasonable starting state (for example using SKM or any of the many HMM algorithms implemented in QuB, SPARTAN, or SMART)? This is important to justify the method being widely valuable, rather than just an incrementally better change-point algorithm.

7) The manuscript is lacking algorithm and implementation details. For example, the authors claim to use the Viterbi algorithm for idealization of time traces using the output of DISC, but the details of this step are never described. Looking at the code, it appears that the authors actually implemented the segmental k-means algorithm (SKM; Qin, 2004) using the state sequence from DISC as the input for the first iteration. SKM iteratively runs the Viterbi algorithm to identify the optimal state sequence followed by re-estimation of model parameters using this state sequence. Strictly speaking DISC does use Viterbi, but the description in the manuscript is not as novel as would appear. This is a critical point because it suggests that DISC just implements an existing, widely-used approach (SKM), and only differs in that it provides a means to initialize the starting model. If this is the case, it should simple be stated in this way. If this is not the case, then further clarity is definitely needed.

8) The SNR of the simulated traces in Figure 2A and Figure 3B seem to be far higher than specified. The most meaningful SNR, as it's usually defined, is the fluorescence intensity relative to the standard deviation of fluorescence background noise (or noise within the signal, if it is static). Correspondingly, SNR is a measure of experimental noise and is independent of the interpretation of the data. The authors instead define SNR in an unusual way that measures the noise in the signal relative to the separation between states (see Equation 2 in the Materials and methods). This may be a valuable metric, but it should not be called SNR. This is very misleading for the field. As used, it is also problematic because it has a circular dependence on the interpretation of the data.

9) In the system that was used to demonstrate the power of the DISC method the authors select for high SNR data and they know that there are five states. The question they sought to answer was whether or not the data exhibited random or cooperative transitions between states, hence, it is not clear that the data even needed to be idealized to assess this question. Gaussian fitting can be used to define the number of states – particularly when the SNR is this high; the idealization thus only provides access to kinetic information, which the authors do not appear as interested in. Hence, the biological question that appears to be of interest relates to the question of cooperativity potentially arising from the transmembrane domains, and this would have been a good way to showcase the method, but this assessment is not included.

10) Some important performance benchmarks are not discussed in the manuscript. For example, how well does it deal with changing or drifting baselines? It is a common feature of fluorescence measurements to be affected by steady reduction in the average fluorescence (drift). This is sometimes also encountered in other type of single-molecule experiments. It would be important for the authors to comment on the performance of DISC under such circumstances.

---

## [Author Response]

Essential revisions:The paper needs to be rewritten in a manner that enough formal description of the method is given to allow the target readership (SM experimentalists, presumably) to understand how it works and what exactly it does. As it is, the paper reads more like an "advertisement" of an algorithm described in detail elsewhere.

We have taken this comment seriously and have made dramatic rewrites to the paper, particularity in the describing how the DISC algorithm works. The motivation and mathematics behind the DISC algorithm have been added in a newly written “Theory” section of the manuscript.

1) The new Divisive Segmentation (DS) algorithm is described in general terms in the text but a formal description is not provided. A reasonable description of the CP analysis portion of DS is provided in Supplemental Note 2 (though a suitable pointer to this from the main text is not given). In the text you then note that you employ a modified k-means algorithm. The reader asks, what is the modification? The citations are unhelpful. Finally, the HAC part is also fairly well described in Supplemental Note 3. But the heart of the DS algorithm does not receive a similarly formal description. A similar problem arises with the mention of the Viterbi algorithm. We are told that you "use" it, but the only references are to the ion-channel and speech-recognition literature. Beside not explaining briefly what it does (for the naive reader's benefit), you do not explain what emission probability densities you use, nor how you set transition probabilities.

We thank the reviewers for pointing out these shortcomings in our earlier version. We have added a new Theory section to clarify all aspects of the DISC algorithm. Details from the previous Supplementary Notes 2 and 3 have been added to the main text to elaborate on change-point detection, unsupervised clustering, and the use of the Viterbi algorithm. New text has been added to provide a formal introduction to the novel divisive segmentation algorithm. All modifications are included in the new Theory section.

2) A question that arises regarding assumptions and parameters: how well does the DS/HSC perform for traces with rarely-visited states? Your selection of total fraction bound > 0.5 (subsection “Trace selection and analysis of binding activity”) suggests that your approach might be best for situations where the various states are roughly equi-probable. An important point to check is the performance of DISC as a function of the kinetics of the underlying process. How well does DISC estimates the number of states for fast kinetics, with a small number of data points in each state (short dwell-times)? One of the goals of single-molecule analysis is the determination not only of the most probable number of states, but also the dwell-times in each state and the transition rates between them. Was an effort made to include dwell-time estimation in DISC?

We agree with the reviewers that the ability to detect rarely-visited states and fast dwell times are important features of any idealization algorithm. We would like to clarify that our use of trace selection based on at least 50% bound fraction criteria was intended to ensure that we are analyzing trajectories of functional proteins. This criterion was applied only to the experimental data but not to simulated data traces which were used for benchmarking DISC with other existing idealization algorithms. Therefore, we do not think that the utility of DISC is limited to trajectories containing at least 50% bound fraction.

Our simulations also highlight the ability of DISC to resolve short lived states and fast dwell times. Figure 2 plots event detection vs varying rates across 2 orders of magnitude, wherein the fastest rate of k_off_ = 1 frames^-1^ is present on average ~3% of the time. This figure shows that DISC maintains a high precision and recall for fast rates. Similarly, our multiple ligand binding simulations show, as expected, that the state occupancies for each additional bound state is progressively lower and the state with four ligands appears only 1% of the time. DISC can resolve these states with high accuracy and precision.

To emphasize the changing state occupancies in each simulation, we have added Figure 2—figure supplement 1 and Figure 3—source data 1. Figure 2—figure supplement 1 shows the occupancy of each of the two states with varying k_off_ values and plots accuracy values as a function of state occupancy. Figure 3—source data 1 contains a table that shows the average time spent in each liganded state for each simulated number of CNBDs. In regards to Figure 2—figure supplement 1, the following text has been added:

“Notably, as the changes in rates also affect the change in state occupancy, the high accuracy values returned after Viterbi refinement further highlight the power of DISC for resolving short-lived and rare transitions (Figure 2—figure supplement 1).”

3) The authors need to be careful using the term high throughput as this means different things to different scientific fields. For instance, high throughput drug discovery often means looking at millions of compounds. Certainly 4000 molecules is relatively high throughput compared to a handful, but even the earliest single-molecule fluorescence papers included a hundred traces, so does a 40x increase justify the semantic transition to high throughput?

We have toned down our use of the word “high-throughput” throughout the manuscript. However, we do wish to empathize that our analysis involved an initial 13,670 trajectories, each comprised of 8000 frames, totaling 1 x 10^8^ data points. This dataset is dramatically larger than standard single-molecule experiments. This is not a 40x increase in the number of trajectories, but a >100x increase in the number of trajectories, and with each trajectory recorded for 10x longer than typical single-molecule fluorescence trajectories. Therefore, the total experimental time analyzed was 1,000x larger than the norm. Importantly, other algorithms are insufficient to process data at this scale with accurate results, as we state in our text about this dataset:

“DISC successfully processed this entire data set within 20 minutes using a standard MacBook Air (1.6 GHz Intel Core i5). The same analysis completed with STaSI yielded unphysical results in 4 hours (Figure 5—figure supplement 1). We estimated analysis by vbFRET would take weeks to complete and was therefore not performed.”

4) As authors show in the assessment of simulated traces in Figure 3, none of the methods compared work in a meaningful way when looking at SNR less than or equal to 3. The data simulated in the manuscript appears to be >10:1 according to a first approximation. So none of the methods assessed does a meaningful job revealing the 'true' number of states even when the data appear to be very good.

We agree with the reviewers that this is indeed a surprising result. The simulated data plotted in Figure 3 is of SNR = 6 as indicated in the figure caption. We believe STaSI and vbFRET overfit the number of states because of heterogenous intensities included in our simulations (Figure 3—figure supplements 1-3). Without the inclusion of this additional noise source, the accuracy of each algorithm improves (Figure 3—figure supplement 4); however, this is real noise we observe and its therefore a more accurate representation of the algorithm’s performance on real data, and a major advantage of our new method. The tendency of STaSI to overfit the data is additionally highlighted in Figure 5—figure supplement 1. We discuss this phenomenon in our text:

“For SNR > 3, which accounts for most of our experimentally obtained data (Figure 2—figure supplement 1C), DISC performs exceptionally well with highest average accuracy (0.91 ± 0.05) and is robust against false positives (precision = 0.96 ± 0.04) and false negatives (recall = 0.93 ± 0.03) across all simulated conditions (Figure 3A). While vbFRET matches the recall of DISC in this SNR range (0.94 ± 0.05), the tendency to overfit the number of states at higher SNR lowers precision (0.80 ± 0.18) and overall accuracy (0.76 ± 0.19). We find STaSI returns the lowest overall accuracy (0.47 ± 0.17) likely do to an overfitting the number of states (precision = 0.57 ± 0.2) and a tendency to miss transitions (recall = 0.75 ± 0.10). Notably, DISC is the only method unaffected by inclusion of heterogeneous state intensities of fcAMP likely due to the use of a Gaussian derived BIC for state selection (Figure 3—figure supplement 4).”

For data less than an SNR of 2, we find different reasons for low accuracy. In the case of DISC and STaSI, often the algorithms return a fit of 1 state owing to the high noise level. In the case of vbFRET, we find that the states are better identified, but the events are poorly located. The ability to detect the states does lead to vbFRET having the highest accuracy for SNR = 2. As we state in our manuscript:

“While no algorithm can idealize a trajectory in the presence of SNR = 1, DISC returns the lowest accuracy at SNR = 2. We suspect this result is due to using the robust BIC for state detection and accuracy would likely be improved with less penalizing objective functions, such as AIC. While vbFRET performs the best at SNR = 2, the overall accuracy is still quite low: an average accuracy value for each number of simulated states is near chance. This low value demonstrates the inability of many algorithms to analyze data in presence of high noise and reinforces the common practice of discarding noisy data to create a more reliable data set.”

5) In the Introduction and Discussion sections, the authors downplay the general usefulness of existing Markov modeling methods. This is unfortunately misleading. The value of Markov modeling for time-series data analysis is proven in the extensive literature across a large number of fields using a variety of experimental approaches. While recent advancements in data throughput (e.g., sCMOS cameras) have put pressure on these algorithms to be efficient, this is a relatively minor problem that can be overcome by better implementations. Related to the above, Supplementary Note 1 includes a great deal of important introductory information that does a better job of placing DISC into the context of existing methods. This information belongs in the main text to substantively review the subject matter.

Your point is well-taken. We have rewritten our Introduction and Discussion sections to address these concerns. Specifically, Supplementary Note #1 was removed from the Supplementary Materials and reworked into our Introduction section. We believe the new section paints a fair portrait of the power of HMM for standard single-molecule analysis while also highlighting their current limitations for application to large datasets.

6) Given that the authors know (or should reasonably expect) that their experimental system has exactly five states, how does their "model free" approach compare to a traditional, model-based approach with a reasonable starting state (for example using SKM or any of the many HMM algorithms implemented in QuB, SPARTAN, or SMART)? This is important to justify the method being widely valuable, rather than just an incrementally better change-point algorithm.

We did reasonably expect our system to have 5 states which is one reason we chose this system to showcase our DISC results. By fitting all > 13,000 trajectories with DISC, we see that 89% of the trajectories returned a fit between 1-5 states, which suggests DISC is doing a good job in identifying states of experimental data. The same analysis with STaSI yielded poor results (Figure 5—figure supplement 1).

Importantly, the inclusions of empty ZMWs, protein decay during excitation, and the probability of more than one protein per ZMW led to distribution of observed number of states. Since our data was heterogenous, it needed to be idealized on a trajectory-by-trajectory basis. If every trace was homogenous, we expect other model-building software such as QuB, SPARTAN, or SMART would do a better job, albeit perhaps slower. However, none of these algorithms can perform trajectory-by-trajectory analysis without a user manually building a separate model for each trace, making the analysis of >13,000 unfeasible. It is for this reason we compared DISC against STaSI and vbFRET, which also perform trajectory-by-trajectory idealization. We state our rationale for choosing these algorithms in the manuscript:

“We benchmark the results of DISC against commonly used HMM and CP-HAC methods: vbFRET and STaSI (Bronson et al., 2009; Shuang et al., 2014). These algorithms were chosen following the results of a recent comparative study that determined these to be the best performers among their class of analysis methods (Hadzic et al., 2018). In addition, DISC, STaSI and vbFRET all perform trajectory-by-trajectory idealization and are written entirely in MATLAB (MathWorks) which standardizes computational performance (Materials and methods).”

7) The manuscript is lacking algorithm and implementation details. For example, the authors claim to use the Viterbi algorithm for idealization of time traces using the output of DISC, but the details of this step are never described. Looking at the code, it appears that the authors actually implemented the segmental k-means algorithm (SKM; Qin 2004) using the state sequence from DISC as the input for the first iteration. SKM iteratively runs the Viterbi algorithm to identify the optimal state sequence followed by re-estimation of model parameters using this state sequence. Strictly speaking DISC does use Viterbi, but the description in the manuscript is not as novel as would appear. This is a critical point because it suggests that DISC just implements an existing, widely-used approach (SKM), and only differs in that it provides a means to initialize the starting model. If this is the case, it should simple be stated in this way. If this is not the case, then further clarity is definitely needed.

We agree with the reviewer that our previous version did not draw close enough parallel between DISC and SKM. We have added the following text to our Theory section when discussing the resolving power of the Viterbi algorithm:

“In general, this improvement was anticipated since we are not the first to apply the Viterbi algorithm to the problem of idealization using unsupervised clustering. Although commonly used in the application of HMMs for hard assignment of data-points into *K* discrete states, the SKM algorithm has shown that the Viterbi algorithm can successfully decode a path sequence following state clustering using k-means as opposed to more rigorous HMM training procedures (Juang and Rabiner, 1990; Qin et al., 2004). Therefore, both SKM and DISC can yield the event detection power of standard HMM approaches without the need of rigorous model training. However, unlike SKM, DISC has the added benefit of identifying the states naturally without the need for any user supervision such as initial state specification. This makes DISC a powerful alternative as a computationally efficient unsupervised single-molecule analysis algorithm.”

8) The SNR of the simulated traces in Figure 2A and Figure 3B seem to be far higher than specified. The most meaningful SNR, as it's usually defined, is the fluorescence intensity relative to the standard deviation of fluorescence background noise (or noise within the signal, if it is static). Correspondingly, SNR is a measure of experimental noise and is independent of the interpretation of the data. The authors instead define SNR in an unusual way that measures the noise in the signal relative to the separation between states (see Equation 2 in the Materials and methods). This may be a valuable metric, but it should not be called SNR. This is very misleading for the field. As used, it is also problematic because it has a circular dependence on the interpretation of the data.

We agree with the reviewer that this term is indeed misleading for experimental data. Our goal was to create a metric that accounted for the separation between state intensities while also accounting for the noise. For the experimental traces, we have changed the term in the manuscript to “state separation” since the computed SNR value would indeed be a result of the idealization and not a true SNR. The term SNR was left unchanged for simulations as it is the appropriate term in that context, and uses the traditional definition that the reviewer describes. The Materials and methods section has been updated with the following text:

“To reflect the ability to cleanly resolve the individual occupation states, we computed the separation of each sequential state vs the noise within a state by:

StateSeparation=1K∑i=2K(µi-µi-1)σi-1

where K is the total number of states, μ is the mean intensity value of a state, and σ is the standard deviation of the data points belonging to a state. This ensures that states are separated well enough to resolve, as would be expected for sequential ligand binding. Therefore, traces featuring between 4-6 identified states with state separation ≥ 3 were retained for analysis.”

9) In the system that was used to demonstrate the power of the DISC method the authors select for high SNR data and they know that there are five states. The question they sought to answer was whether or not the data exhibited random or cooperative transitions between states, hence, it is not clear that the data even needed to be idealized to assess this question. Gaussian fitting can be used to define the number of states -particularly when the SNR is this high; the idealization thus only provides access to kinetic information, which the authors do not appear as interested in. Hence, the biological question that appears to be of interest relates to the question of cooperativity potentially arising from the transmembrane domains, and this would have been a good way to showcase the method, but this assessment is not included.

We agree with the reviewers that we did not use our idealization to their full potential. Therefore, we modeled our idealized trajectories in QuB to directly extract global rates for a five-state sequential HMM. The results of this analysis are now shown in Figure 5D,E. This process optimized on and off rates from our idealized trajectories show a linear relationship. This provides further evidence to the non-cooperative nature of our system. The manuscript has been updated to address this new analysis.

10) Some important performance benchmarks are not discussed in the manuscript. For example, how well does it deal with changing or drifting baselines? It is a common feature of fluorescence measurements to be affected by steady reduction in the average fluorescence (drift). This is sometimes also encountered in other type of single-molecule experiments. It would be important for the authors to comment on the performance of DISC under such circumstances.

We acknowledge that not all possible performance benchmarks have been addressed in this manuscript; however, we believe our simulations have thoroughly explored DISC accuracy across a variety of dynamics, number of states, signal-to-noise to be confident in its performance on typical single-molecule data. As with most idealization algorithms, pre-processing the data (e.g. filtering, baseline correction, etc.) prior to idealization will likely increase idealization accuracy. There is nothing inherently built in to the DISC algorithm to be aware of a drifting baseline and therefore DISC accuracy will likely decrease with the magnitude of drift. We have added the following to our Discussion section to ensure the readers are aware of this limitation:

“However, there is no inherent knowledge within DISC to consider varying sources of experimental noise, such as photo-blinking or baseline drift. Therefore, the idealization accuracy of DISC will likely be improved when used after typical pre-processing of single-molecule trajectories.”

[Editors' note: we include below the reviews that the authors received from another journal, along with the authors’ responses.]

We thank the reviewers for their continued feedback and critical assessment of our work. We have addressed the provided comments and, in the process, significantly strengthened the manuscript. Our new changes are specifically aimed to more concisely explain our approach and provide additional context for the use of DISC to a more general audience.

Reviewer #1:1) The revised manuscript has addressed my concerns regarding generality of the approach and improved presentation. The manuscript is now acceptable for publication in Nature Communications.

We thank the reviewer for their kind words and thoughtful comments that have helped strengthen our manuscript.

Reviewer #2:I have reviewed this paper before. While the authors have addressed minor comments, the main difficulty lies with the heart of the paper itself.Let me explain briefly again the heart of the problem. The Hidden Markov model (HMM) has been a paradigm shifting model across fields because, given a noise model and the assumption of Markov transitions between states, it finds: 1) noise parameters; 2) transition matrices; 3) trajectories consistent with those assumptions.Because the HMM has a coupled kinetic and observation model (which is supported by Mathematics which are 100% clear), as we apply the HMM to data, it will sometimes adjust the noise level in one state to make sure that the Markovian (exponential waiting time) assumption is not violated overall.Now the authors do not use the HMM as, in any HMM, the number of states must be specified. Fine.Suppose, for argument’s sake, that the number of states was known and, for whatever reason, we chose not to use a (normal) HMM because perhaps the number of states was too large and the forward filter was too slow (e.g., too long a time trace). Now, suppose, as the authors do, that the data were to be messy, and we were first to denoise traces before learning kinetics. Then what?The short answer is that any denoising/clustering step that were not simultaneously consistent with the Markov assumption and the noise model would give results that are then inconsistent with our starting assumptions. In other words, garbage in, garbage out.In the case of the current paper, what I highlight here is just the very beginning (the divisive segmentation step) of the very many steps that run against the basic assumptions of the (unknown state) HMM that forms the heart of their problem.Indeed, for the current work, the problems are much worse as the number of states themselves are unknown.I understand the need, as the authors do, for an efficient state number identifier within the HMM paradigm. I think there are many approximate (or even deterministic/non-Bayesian) ways forward (perhaps a Laplacian approximation on a likelihood or an ABC within a Bayesian paradigm) that could be conceived of. But the math needs to make sense. Right now, the math makes no sense.I have no doubt that the method proposed by the authors is efficient at what it does. But I would never use it, nor would I recommend anyone use it because the steps in the algorithm are mutually contradictory and the method (as captured in Figure 1 of the supplement) is just plain wrong.

While we accept the criticism that our method is not based on completely self-consistent first principle arguments (e.g. theoretically optimal scenario), we strongly assert that this tradeoff is completely acceptable given the significant consequent gains in performance (e.g. practicality and efficiency). In fact, the tradeoff for functionality over optimality in analysis is crucial for scientific progress. Consider super-resolution localization microscopy wherein in single emitters are identified beyond the diffraction limit by fitting to the point-spread function. Although the theoretically optimal scenario is to fit each emitter using a 2D Bessel Function, the entire community fits with 2D Gaussians. Not only would the more rigorous fitting be impractically slow, but no additional benefit can be derived by fitting with 2D Bessel functions under the relevant experimental conditions (finite noise, etc.). The use of 2D Gaussians is not as mathematically rigorous, but it is incredibly practical. Similarly, the semi-empirical force fields that make molecular dynamics simulations so accessible and powerful are not as rigorous as ab initio methods, but again, the community of users is fine with the compromise: a negligible degree of accuracy is sacrificed for a tremendous benefit in practicality. If anything, these less than optimal approaches are much more prevalent, by a wide margin, than the theoretically self-consistent methods.

In the case of DISC, we show the value of practical applicability provides tremendous and enabling advantages for large data sets. The fact that accuracy does not suffer in our method is direct evidence that the sacrifice of first principles rigor is miniscule and completely acceptable. In fact, the common tradeoffs that we found were being made in the larger big-data communities served as the inspiration for this work. For example, although reviewer 2 has clear issues with our use of a hybrid change-point and clustering algorithm for determining states in a trajectory, similar divisive clustering approaches minimizing “ad-hoc” objective-functions are extremely common in statistics (e.g. X-Means^1^ [2,641 citations] and Bisecting K-Means^2^ [3,157 citations]) and companies such as Twitter rely on change-point detection algorithms daily to analyze production and user experience data^3^. Further, the segmental k-means algorithm (SKM, 564 citations)^4^ assigns data points into a set number of user-provided states using k-means clustering and uses the Viterbi algorithm to find events. Pioneering software such as QuB for analyzing single-ion channel trajectories heavily depend on the SKM algorithm owning to its computational speed and accuracy (202 primary citations)^5^. Indeed, a recent groundbreaking study in Nature Methods described the use of SKM to analyze their large dataset of >10,000 molecules^6^. These authors make it abundantly clear that:

“State assignment is performed using SKM. Although several other Markov modeling utilities are available and could be used for this purpose (e.g., HaMMy, vbFRET, SMART, iSMS and TwoTone), SKM is much faster than the alternatives, and this facilitates analysis of the large data sets associated with sCMOS.”

Experimental labs on the forefront of technological breakthroughs need more efficient algorithms to keep up with the growing scale of data generation. Although some degree of efficiency can be achieved by more sophisticated hardware (as discussed below by reviewer 3), this by no means diminishes the value of new algorithms capable of performing high throughput analysis on standard computers. We believe our manuscript demonstrates that DISC offers dramatically superior speed with maintained accuracy for both simulated and real data sets. We expect the majority of biophysical investigators will have no issue trading theoretical optimality for practical applicability.Many groups have already contacted us about using DISC, providing conspicuous evidence of this prediction.

Also, in regards to Supplemental Figure 1, to our knowledge this figure is correct. However, if something is indeed wrong, we would appreciate a reviewer to tell what is wrong (and why) to strengthen our manuscript and ensure we properly convey information.

Reviewer #3:Authors have done a good job with providing new material to answer the initial questions and technical/experimental concerns of this reviewer and in my opinion this article is now almost ready to showcase the DISC method in a fair manner. My main concern, that was brought upon with the authors before and here again to a lesser degree would remain, is the way information is presented for the most probable group of users: biologists. Since the goal in publishing any scientific paper is eventually to benefit the researchers who would want to implement and take advantage of the knowledge or techniques presented, I would suggest authors to name multiple real-world biology related application examples in the text so a user that doesn’t carry much technical knowledge in analysis techniques and machine learning, would be able to grasp the limitations and applications for their own class of projects. That said, my opinion about the scope and presentation of DISC is positive and would like to see it in print. There are still few areas that need attention and I will list them below. Out of the fourteen points this reviewer brought up with the authors before, nine are resolved and five need attention. Four of these five points will be resolved by adding a paragraph or changing parts of statements and more clarification/specification in the paper. The last one needs a new run of the simulation. Other than that, there are two suggestions with respect to figures that need to be addressed properly. To briefly point these out: Comments#1, #2, #5, and #6 can be resolved by adding/altering text in the article while comment#9 is of technical nature and needs to be resolved by plotting the requested information. With respect to Figures, Figure 1 needs details for the axis and Figure 4 needs a new run of simulation to present a vital information for the limiting case of lack of detection. Details come below.Note: "Comment#N" is a comment to the provided answer by the authors to question #N in the former round of review by this reviewer.

We appreciate the thorough and thoughtful comments from this reviewer which have helped us strengthen our manuscript. We agree that the tone of our manuscript and the reliance on technical jargon (e.g. unsupervised clustering) is not friendly to non-expert audiences that will be the main users of our technology. As a result, a large portion of our “The DISC Algorithm” section has been rewritten to reduce unfamiliar jargon and make the assumptions and limitations of DISC clearer.

1) Comment#1: Including the reference and mentioning it in the text satisfies the question. A small point would be, the speed is not a main issue since high throughput data analysis can always get addressed with the introduction of new better hardware which happens every year.

We thank the reviewer for pointing this out. Although technological progress will certainly continue to improve computational speed, faster algorithms benefit everyone now rather than sometime in the future, and also benefit many labs that do not consistently have the very latest hardware or experience to build the optimal hardware platform.

2) Comment#2: although the answer provided is not convincing, the added part in the paper is good enough to cover this issue. The reason the answer is not convincing is: hardware advances happen every year and within a couple of years, there will be more options for faster analysis and on the other hand, in many labs, working with GPU and parallel processing is becoming routine and a computer science/electrical engineering/physics PhD student or postdoc can implement methods to achieve shorter analysis times this way.

See response to point #1 about hardware and other technological advances.

3) Comment#5: what the word “unsupervised” bears is on dispute here. The fact that after picking a function based on the parameters of the system and the hardware you don’t need supervision, doesn’t remove the need for picking that function and make sure that it is the correct function. That said, unsupervised learning has a specific meaning used in analysis which is in line with what authors include. So the issue would be mainly explaining this clearly for the readers. Author’s usage of the technical term unsupervised learning might be misleading for the reader who is not familiar with machine learning terminology and in search of a method for data analysis. So maybe authors want to rephrase their description to clearly state that the unsupervised analysis doesn’t mean that the user can’t implement the code in a wrong way and achieve results that are not trustable, but the user can only trust the results if the proper function is known and implemented initially. The fact that they took it from supplementary note 5 to the main text, is good and removes the initial apparent contradiction in their text. As long as the discussed point is stated clearly in the main text, the issue is resolved in the opinion of this reviewer.

We agree with the reviewer that our language was unclear and potentially misleading. We have therefore made corrections throughout the manuscript when appropriate to ensure the definition of “unsupervised” is conveyed and discuss exactly what the user has to do to provide a reasonable answer. We have also added the following text to the final paragraph of our “The DISC Algorithm” section to highlight the importance of implementing DISC correctly:

“However, it is important to note that while DISC performs idealization using unsupervised clustering, obtaining accurate results does require the user to determine the appropriate information criterion and change-point detection methods to use as obtained results heavily depend on these variables.”

4) Comment#6: A satisfying answer to this question would take into account different type of problems a biologist would be able to investigate with DISC method and give examples of real systems and problems. Authors have done this to some extent in discussion but giving real biological examples to show the abilities and limitations would always help. I clarify that this is not a suggestion for the authors to perform biological experiments but to showcase and discuss possible and specific applications and why they can benefit or not by this method.

We agree with the reviewer direct applications of DISC to specific biological problems was poorly showcased. We have modified our Discussion section to include the new text.

“Finally, our results show that DISC provides a dramatic improvement in computational speed over current state-of-the-art approaches while either improving or maintaining high accuracy for both state determination and event detection. This speed increase is directly applicable to analyzing the growing data-sets obtained in single-molecule fluorescence paradigms to adequately sample population dynamics. For example, the use of sCMOS cameras enabled smFRET measures of tRNA conformational changes during protein translations across thousands of molecules simultaneously with millisecond resolution^7^. Additionally, magnetic tweezers have enabled week-long mechanical measurements of single-protein folding and misfolding, shifting observable dynamics to pathological timescales and allowing the detection of incredibly rare events(Popa et al., 2016).”

5) Comment#9: Based on the answer authors prepared to this point, this question rises: if two points are closer than the detection ability of the method, the method wouldn’t be able to resolve them so the precision would fall, or as the authors rephrased in their answer to the review, the signal will reduce to noise (this case is not showcased in the color map in Figure 3.) meaning no detection happened for at least one of the points (since two or more points are seen as one) so since authors answered this question with a description related to noise level, such a result is not seen in the color panel since we see only signal to noise levels of above 3. It would be useful if the authors write a paragraph about this issue (meaning about the limitation of detection in too dense areas or too noisy areas) and then bring up their interpretation that states in such a case they interpret the unseen emitter as noise which then needs to be shown in their figure as signal to noise ratios of 1 and 2 (currently it shows 3 and above). This will clarify the resolution limit question the reviewer had before.

We remind the reviewer that we are looking at a *signal-to-noise ratio*, and not signal and/or noise separately. We therefore believe our simulations have addressed the valid concern of the reviewer, albeit in a different manner.

We compute signal-to-noise (SNR) as the height between states divided by the standard deviation of the data points in the previous state (as described in our Materials and methods). For example, if we have a two-state system with observed intensities of state [1] = 100 and state [2] = 200, then we have a step height of 100. If we apply Gaussian noise to this process at σ [1] = 20, then SNR = (200100) / 20 = 5. Now, if we were to only change the height of state [2] as the reviewer has suggested, say to state [2] = 150, then we would have: SNR = (150-100) / 20 = 2.5. Therefore, changing the step height without changing the σ [1] results in a changing of SNR. However, if we were to change σ [1], say to σ [1] = 10, we would have a SNR = (150-100) / 10 = 5. Thus, detecting a step height detection completely dependent on the noise in the trajectory and is not an independent entity.

Also, detecting a minimal step height is less translatable than assessing SNR ratios given that different experiments have different expected observed values. For example, while our EMCCD experiments obtain intensities values on the scale of 100s arbitrary units, correlative FRET experiments [e.g. smFRET] only has a range of 0 to 1.

In regards to Figure 3, we have updated the figure to include simulations are SNR = 1 and 2. The text has also been updated to address this change.

“Across all the simulations, DISC provides the highest average accuracy, precision and recall (Figure 3A). While no algorithm can idealize a trajectory in the presence of SNR = 1, DISC returns the lowest accuracy at SNR = 2 as result of using the robust BIC for state detection and would likely be improved with less penalizing objective functions, such as AIC. While vbFRET performs the best at SNR = 2, an average accuracy value for each number of simulated states is close to chance. This demonstrates the inability of many algorithms to analyze data in presence of high noise and reinforces the common practice of discarding noisy data to create a more reliable data set. For SNR > 3, which accounts for the majority of our experimentally obtained data (Supplemental Figure 2), DISC performs exceptionally well with highest average accuracy (0.91 ± 0.05) and is robust against false positives (precision = 0.96 ± 0.04) and false negatives (recall = 0.93 ± 0.03) across all simulated conditions (Figure 3A). While vbFRET matches the recall of DISC in this SNR range (0.93 ± 0.05), the tendency to overfit the number of states at higher SNR lowers precision (0.80 ± 0.18) and overall accuracy (0.76 ± 0.19) (Figure 3A). Notably, DISC is the only method unaffected by inclusion of heterogenous state intensities of fcAMP likely due to the use of a Gaussian derived BIC for state selection (Supplemental Figure 8).”

6) Figure 1: sub figures a and b need specific indication for intensity and time so we know what is the time period we are looking at and if the intensity is normalized or not.

The goal of Figure 1 is only to convey how the algorithm works on an example trajectory and therefore have removed details for clarity (e.g. only showing 5 date points the Viterbi Trellis of Figure 1A and not showing BIC in Figure 1C). As we show the results of DISC on plenty of figures in the main text and supplemental, we prefer not to update this figure with additional details simply to reduce clutter.

7) Figure 4: It’s necessary that this plot includes the case for signal to noise ratio of 1. This would be a good test to see if the method is working fine in the limiting case where emitters are not recognized. If it provides an unreasonable accuracy at signal to noise ratio of 1, there would be a flaw in the model that needs to be taken care of before other people start to use the code. This is particularly important since in practical cases, samples might provide a range of signal to noise ratio in different locations even in a single experiment, so the user wouldn’t want to mistakenly take noise for data specifically if the method is running unsupervised and provides wrongly high accuracy for such situations. The current plot doesn’t investigate this possibility and takes it for granted.

We agree that showing that DISC will indeed fail in a scenario of low SNR is important to assess the reliability of the method. We have updated Figure 4 with a new simulation showing accuracy results of DISC at a SNR = 1 across varying number of data points. As expected, DISC performs poorly at this signal-to-noise ratio.

References

1) Pelleg, D. & Moore, A. W. in *Icml.* 727-734.

2) Steinbach, M., Karypis, G. & Kumar, V. in *KDD workshop on text mining.* 525-526 (Boston).

3) James, N. A., Kejariwal, A. & Matteson, D. S. Leveraging Cloud Data to Mitigate User Experience from 'Breaking Bad'. *2016 Ieee International Conference on Big Data (Big Data)*, 3499-3508 (2016).

4) Juang, B.-H. & Rabiner, L. R. The segmental K-means algorithm for estimating parameters of hidden Markov models. *IEEE Transactions on acoustics, speech, and signal Processing***38**, 1639-1641 (1990).

5) Qin, F. Restoration of single-channel currents using the segmental k-means method based on hidden Markov modeling. *Biophysical Journal***86**, 1488-1501, doi:10.1016/s00063495(04)74217-4 (2004).

6) Juette, M. F.et al. Single-molecule imaging of non-equilibrium molecular ensembles on the millisecond timescale. *Nature Methods***13**, 341-344, doi:10.1038/nmeth.3769 (2016).